# Bridging Efficiency and Adaptability: Continual Learning of MLPs on Class-Incremental Graphs

**Qiannan Zhang**                                                    *qiz4005@med.cornell.edu*
*Cornell University*

**Shichao Pei**[*]                                                   *shichao.pei@umb.edu*
*University of Massachusetts Boston*

**Reviewed on OpenReview:** *https:// openreview. net/ forum? id=3KYwHaKXcn*

## Abstract

Compared to static graphs, class-incremental graphs place higher demands on inference latency to support timely predictions for newly emerged node classes, especially in latency-sensitive applications. However, the high inference cost of Graph Neural Networks (GNNs) limits their scalability and motivates GNN-to-MLP distillation, which transfers knowledge from a GNN to a Multi-Layer Perceptron (MLP) to enable graph-free, low-latency inference. Yet, existing efforts focus on static graphs. When directly applied to class-incremental graphs, they inevitably suffer from the high computational cost of frequent GNN updates and MLP's inability to retain knowledge of previously learned classes. To bridge efficiency and adaptability, we propose a novel framework featuring an asynchronous update paradigm between GNN and MLPs, allowing rapid adaptation to evolving data. MLPs employ a progressive expansion strategy for continual adaptation and an energy-based routing mechanism for test-time inference. During GNN updates, knowledge from MLPs trained in the current cycle is distilled back into GNN to preserve long-term knowledge. Experiments on real-world datasets demonstrate that our framework achieves superior performance on class-incremental graphs, effectively balancing adaptability to new data and inference efficiency.

## 1 Introduction

Graph Neural Networks (GNNs) (Veličković et al., 2018a; Kipf & Welling, 2017) have demonstrated remarkable success in modeling graph-structured data by leveraging message passing mechanisms (Wu et al., 2020) to capture relational dependencies. However, the reliance on message passing incurs significant inference latency due to recursive aggregation (Zhang et al., 2022b). This limits the scalability of GNNs, making them impractical for large-scale applications where rapid inferences are essential, such as real-time recommendation systems (Huang et al., 2025) and online search (Zhang et al., 2022a). To address this, GNN-to-MLP distillation (Zhang et al., 2022b; Wang et al., 2023) has emerged as an effective solution, transferring knowledge from a GNN to a graph-free Multi-Layer Perceptron (MLP), as illustrated in Figure 1A(a). By replacing GNN-based inference with an MLP, this approach enables efficient, low-latency predictions without requiring access to the original graph structure.

While GNN-to-MLP distillation has been widely explored in *static* settings, real-world applications often involve graph class-incremental scenarios (Zhang et al., 2024b; Tan et al., 2022; Lu et al., 2022), where new node classes continuously emerge as the graph evolves. The adoption of GNN-to-MLP distillation in such scenarios remains unexplored. To enable low-latency inference on class-incremental graphs, a straightforward approach is to frequently retrain the GNN and distill it into an MLP at each time step, as Figure 1A(b) shows. However, frequent GNN updates are computationally expensive, as message passing over large-scale graphs incurs significant overhead, making fast adaptation impractical. Meanwhile, the deployed MLP remains

---
[*]Corresponding author.

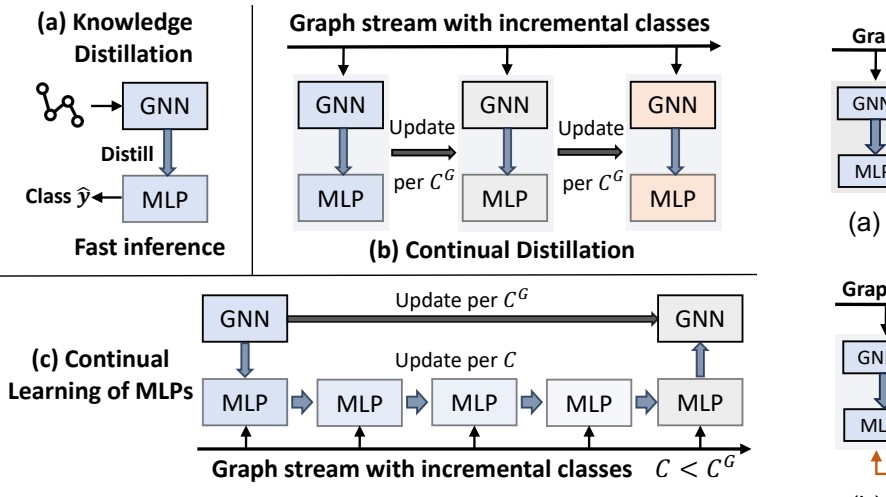

A. Conceptual comparison of (a) GNN-to-MLP distillation, (b) continual GNN-to-MLP distillation, (c) the proposed continual learning of MLPs. $C^G$ and $C$ denote the update cycle for the GNN and MLP, respectively.

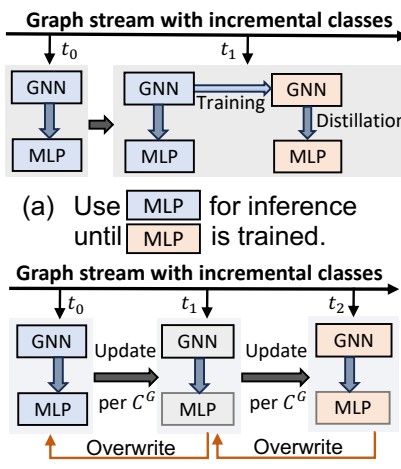

B. Two issues of continual distillation.

Figure 1: Overview of the motivations and challenges in continual learning of MLPs on class-incremental graphs.

static until the ongoing GNN update is completed, failing to rapidly produce reliable inferences on newly emerging node classes. For instance, as shown in Figure 1B(a), when nodes with new classes emerge at $t_1$, the GNN will be updated to adapt to these new classes. Assuming the update takes one day, the previous MLP, which has not been exposed to the new data, remains in use for inference during this period. As a result, it cannot provide accurate predictions for new classes, making inference on the new data infeasible.

Although recent methods (Zhang et al., 2024b; Liu et al., 2023) based on continual learning have been explored in the graph domain to adapt to new data and avoid retraining GNNs from scratch, these methods still focus on incrementally updating a *single GNN model* and have to rely on isolated distillation at each time step when pursuing low-latency inference, preventing MLPs from leveraging previously learned knowledge. Since each new MLP is initialized with the parameters of the previous one but undergoes significant updates during optimization, the retained knowledge is often overwritten, as illustrated in Figure 1B(b), which leads to catastrophic forgetting and inefficient knowledge transfer, hampering generalization across previously seen node classes.

These challenges highlight the need for a framework that bridges the gap between *efficiency* and *adaptability*. The framework should enable low-latency MLPs to remain effective on class-incremental graphs, while mitigating catastrophic forgetting and preserving fast and accurate inference. To enable rapid inference on newly emerging nodes and reduce the high computational cost associated with GNN updates, it is impractical to frequently retrain or continually update the GNN model. Instead, to ensure that the MLP adapts to new graph data in a timely manner, it should be updated more frequently than GNN, as illustrated in Figure 1A(c). This allows the MLP to directly learn from newly emerging nodes and classes and provide accurate inference, while GNN is updated at a lower frequency to reduce computational costs without sacrificing long-term structural learning. However, as the MLP is continually updated at a high frequency, it risks forgetting earlier learned knowledge, leading to inference performance degradation on previously seen node classes. Without an effective mechanism for knowledge retention, the inference model tends to prioritize newly arriving data at the cost of past information. Existing methods for mitigating forgetting face limitations in GNN-to-MLP distillation. Replay-based approaches (Zhou & Cao, 2021; Liu et al., 2023) require identifying which past data to retain, while regularization-based methods (Kirkpatrick et al., 2017; Zhang et al., 2022d) increase complexity and computational cost, often causing class representation drift. The

Table 1: Model Comparisons

| Model Types | Efficiency | Adaptability |
|---|---|---|
| GNN | ✗ | ✗ |
| GNN-to-MLP | ✓ | ✗ |
| Graph Incremental Learning | ✗ | ✓ |
| **CMLP (Our Model)** | ✓ | ✓ |

necessity of maintaining low-latency inference for MLP imposes a strict constraint on designing an effective MLP learning strategy on class-incremental graphs.

**Our work**. To develop a solution that enables the MLP to effectively retain past class knowledge while still rapidly adapting to newly emerging graph data with new classes, all while maintaining low-latency inference, in this paper, we propose a framework called **CMLP**, built on a novel paradigm of asynchronously updating the GNN and MLPs. To avoid complex designs that could increase inference latency, we propose a simple yet effective progressive MLP expansion strategy to address the challenges of graph class-incremental learning while maintaining low-latency inference. The progressively enlarged MLPs preserve both past and current knowledge without overwriting earlier model parameters. As time progresses and MLPs from subsequent stages are trained, rather than treating them as independent models, we ensemble them as a mixture of experts, where each MLP expert specializes in different learning phases, adapting to distinct emerging classes. To effectively utilize these experts during inference, we introduce an energy-based test-time routing mechanism that selects the most suitable expert for a given input node. Unlike traditional gating mechanisms (Zhou et al., 2022) that require training, our approach allows expert selection to be guided by the energy-based confidence scores (LeCun et al., 2007) during test time. Furthermore, these learned experts can serve as teacher models for distillation back into the GNN during its update phase, helping to mitigate the forgetting issue that occurs as the GNN continuously adapts to new data and ensuring a more robust and efficient continual learning process. To summarize, our main contributions are as follows:

- To the best of our knowledge, we are the first to investigate GNN-to-MLP distillation on class-incremental graphs, addressing the challenge of adapting to evolving graph data while maintaining inference efficiency.

- We design an asynchronous update paradigm for GNN and MLPs to enable rapid adaptation to evolving graph data.

- We propose a progressive MLP expansion strategy to mitigate forgetting, ensure low-latency inference, and support fast updates without access to past data.

- We formulate the expanded MLPs as a mixture of experts to leverage knowledge from different stages and introduce an energy-based test-time routing mechanism to efficiently select the best expert for inference.

- We conduct extensive experiments on real-world graph datasets, demonstrating that our method effectively balances adaptability to new data and inference efficiency.

## 2 Related Work

**GNN-to-MLP Distillation.** It has emerged as a promising approach (Zhang et al., 2022b; Wang et al., 2023) to enable graph-free inference by training a lightweight MLP to mimic the predictions of a GNN. This paradigm is particularly appealing in scenarios where low-latency inference is critical. GLNN (Zhang et al., 2022b) pioneered distilling GNN outputs into structure-independent MLPs. Follow-up work (Wang et al., 2023; Tian et al., 2022; Yang et al., 2021) explored injecting structural biases during distillation, while others (Xiao et al.; Dong et al., 2022; Hu et al., 2021; Liu et al., 2022) focused on aligning MLPs with GNNs at the subgraph level. Recent advances include structural discretization (Yang et al., 2024) and ensemble-based frameworks (Lu et al., 2024). Despite these advances, existing methods are confined to static graphs. In contrast, real-world graphs are dynamic in nature with new nodes and classes emerging continuously, requiring models that enable fast inference and continual adaptation without forgetting. However, current

approaches overlook the challenge of combining efficient inference with adaptability in the class-incremental graph setting.

**Graph Class-incremental Learning.** It extends traditional continual learning to graphs, where models must learn from emerging classes while avoiding catastrophic forgetting. Existing methods mainly follow three lines: regularization-based approaches constrain parameter updates to retain prior knowledge (Li & Hoiem, 2017; Kirkpatrick et al., 2017; Liu et al., 2021); memory-based methods replay stored or synthesized data (Zhou & Cao, 2021; Zhang et al., 2022e; Liu et al., 2023); and architectural methods dynamically expand models to handle new tasks (Zhang et al., 2022d; Niu et al., 2024). Despite these advances, existing methods focus on updating a single GNN model, which differs significantly from our problem setting. Applying GNN-to-MLP distillation on class-incremental graph remains unexplored.

## 3    Preliminaries

**Definition 1 (Graph Class-Incremental Learning)** *In graph class-incremental learning, a model learns from an evolving sequence of graph data in which new node classes are introduced over time. Let the initial training graph be $\mathcal{G}_0 = (\mathcal{V}_0, \mathcal{E}_0, \mathcal{X}_0, \mathcal{Y}_0)$, where $\mathcal{V}_0$ is the node set, $\mathcal{E}_0$ is the edge set, $\mathcal{X}_0 \in \mathbb{R}^{|\mathcal{V}_0| \times d}$ is the node feature matrix, and $\mathcal{Y}_0$ is the label set. After the initial phase, the model encounters a sequence of incremental graph data $\{\mathcal{G}_t = (\mathcal{V}_t, \mathcal{E}_t, \mathcal{X}_t, \mathcal{Y}_t)\}_{t=1}^T$, where each $\mathcal{G}_t$ contains data associated with a newly introduced, class-disjoint label set $\mathcal{Y}_t$, i.e., $\mathcal{Y}_t \cap \mathcal{Y}_{t'} = \emptyset$ for $t \neq t'$. Here, $\mathcal{G}_t$ and $\mathcal{G}_{t+1}$ denote consecutive increments in the evolving graph stream, and we do not require a graph-expansion relation such as $\mathcal{V}_t \subseteq \mathcal{V}_{t+1}$ or $\mathcal{E}_t \subseteq \mathcal{E}_{t+1}$. In each phase $t$, the model is updated using only the current incremental graph $\mathcal{G}_t$, while it is required to retain knowledge acquired from previous phases $\mathcal{G}_0, \ldots, \mathcal{G}_{t-1}$ without accessing the raw graph data from any earlier step $t' < t$.*

**Definition 2 (Update Cycles in Incremental Learning)** *The update cycle determines how frequently models adjust to newly arriving data while maintaining efficiency and performance. Let $C^G$ and $C$ denote the update cycles for the GNN and MLP, respectively.*

**Definition 3 (Problem Definition)** *We study the problem of GNN-to-MLP distillation on class-incremental graphs, where the inference model incrementally learns new node classes while ensuring efficient inference and effective adaptation. Given a sequence of evolving graphs $\mathcal{G}_0, \mathcal{G}_1, ..., \mathcal{G}_T$, where each graph is represented as $\mathcal{G}_t = (\mathcal{V}_t, \mathcal{E}_t, \mathcal{X}_t, \mathcal{Y}_t)$, our goal is to incrementally learn MLPs that can classify both past and newly introduced classes while maintaining low inference latency. Formally, given an evolving dataset $\{\mathcal{G}_t\}_{t=0}^T$, $C^G = T$, and $C = 1$, the objective is to learn a sequence of inference models $\{f_{MLP}^t\}_{t=0}^T$ that adapt to new class distributions while preserving knowledge from previous graphs. The problem is constrained by the need for graph-free inference, continual adaptation, and knowledge retention, making it distinct from conventional static graph learning approaches.*

## 4    The Proposed Method

This section details the components of the proposed **CMLP** framework. Figure 2 provides an overview of CMLP. The optimization process is discussed in Appendix C.

### 4.1    GNN-to-MLP Distillation

At the initial time step $t_0$, given a graph $\mathcal{G}_0 = (\mathcal{V}_0, \mathcal{E}_0, \mathcal{X}_0, \mathcal{Y}_0)$, a GNN model $f_{\text{GNN}}(\cdot)$ is trained to learn structure-aware node embeddings as $H_0 = f_{\text{GNN}}(\mathcal{G}_0)$, where $H_0$ denotes the representations of nodes learned by the GNN. The goal of GNN-to-MLP distillation is to transfer knowledge into an MLP model $f_{\text{MLP}}(\cdot)$, enabling it to approximate $f_{\text{GNN}}(\cdot)$ without requiring explicit graph connectivity. The MLP takes as input the node feature matrix $\mathcal{X}_0$ and maps it to the embedding space by $\hat{H}_0 = f_{\text{MLP}}(\mathcal{X}_0)$, where $\hat{H}_0$ represents the node embeddings produced by the MLP using only the feature matrix $\mathcal{X}_0$. A prediction head $g_0(\cdot)$ is then applied to map embeddings to class logits by $\hat{\mathcal{Y}}_0 = g_0(\hat{H}_0)$. To ensure that the distilled MLP effectively

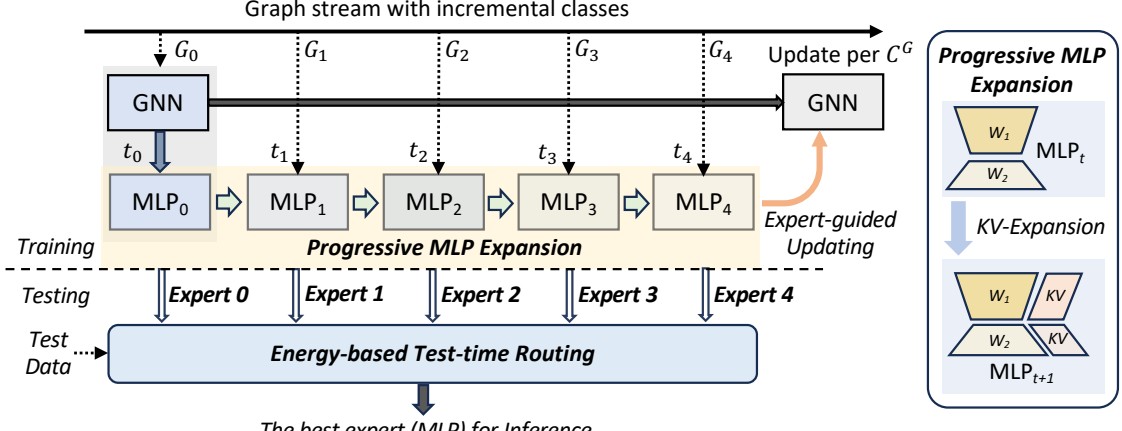

Figure 2: Overview of the CMLP framework. During the training stage, progressive MLP expansion enlarges MLPs through key-value expansion over time, enabling them to learn from newly emerging classes. When the GNN update cycle is reached, GNN incorporates both the data from the current cycle and the guidance of experts for updating. During the testing stage, an energy-based test-time routing mechanism selects the most suitable expert for inference.

mimics the pre-trained GNN, we optimize the following knowledge distillation loss function defined as:

$$\mathcal{L}_{\text{KD}} = \mathcal{L}_{\text{CE}}(\hat{\mathcal{Y}}_0, \mathcal{Y}_0) + \alpha \mathcal{L}_{\text{KL}}(\hat{H}_0, H_0) \tag{1}$$

where $\mathcal{L}_{\text{CE}}$ is the cross-entropy loss ensuring supervised classification performance, $\mathcal{L}_{\text{KL}}$ is the KL-divergence loss aligning the representations learned by the MLP with those of the GNN, and $\alpha$ is a tradeoff hyperparameter. Once trained, the distilled MLP enables graph-free inference, classifying nodes without requiring access to $\mathcal{E}_0$. Note that our work focuses on continual learning of MLPs for class-incremental graphs rather than designing advanced GNN-to-MLP distillation strategies. Any GNN-to-MLP distillation strategy can be adopted and we leverage the canonical GNN-to-MLP distillation approach (Zhang et al., 2022b).

## 4.2 Progressive MLP Expansion

Since class-incremental graphs involve a sequence of graphs $\{\mathcal{G}_t\}_{t=0}^{T}$, the MLP should be continually updated as new graphs emerge to quickly adapt to the emerging graph and ensure rapid and accurate inference. To accumulate new knowledge while minimizing interference with previously learned information, and to ensure that the MLP remains efficient for inference, we design a progressive expansion strategy with a key-value expansion mechanism. Rather than introducing complex learning components that could increase inference latency, this approach incrementally expands the MLP without altering previously learned parameters, also maintaining a lightweight structure suitable for real-time large-scale applications.

Specifically, at each time step $t$, the MLP consists of two layers. Formally, the forward propagation of the MLP at time $t$ is given by:

$$H_t^{(1)} = \sigma(\mathcal{X}_t W_1^{(t)} + b_1^{(t)}), \quad W_1^{(t)} \in \mathbb{R}^{d \times h}, \quad b_1^{(t)} \in \mathbb{R}^h \tag{2}$$

$$H_t^{(2)} = \sigma(H_t^{(1)} W_2^{(t)} + b_2^{(t)}), \quad W_2^{(t)} \in \mathbb{R}^{h \times h'}, \quad b_2^{(t)} \in \mathbb{R}^{h'} \tag{3}$$

where $\mathcal{X}_t$ denotes the feature matrix, and $W_1^{(t)}, W_2^{(t)}$ are the layer-wise weight matrices, and $b_1^{(t)}, b_2^{(t)}$ are bias vectors. $\sigma(\cdot)$ represents the activation function.

At time step $t + 1$, the first-layer weight matrix is expanded to accommodate new feature representations. Instead of overwriting or directly appending weights, we introduce a key-value expansion mechanism with a trainable key $K_1^{(t)}$ and a value $V_1^{(t)}$, which dynamically generates additional weight components by:

$$W_1^{(t+1)} = \begin{bmatrix} W_1^{(t)} & K_1^{(t)} V_1^{(t)} \end{bmatrix}, K_1^{(t)} \in \mathbb{R}^{d \times k}, \quad V_1^{(t)} \in \mathbb{R}^{k \times \Delta h} \tag{4}$$

$$b_1^{(t+1)} = \begin{bmatrix} b_1^{(t)} \\ b_1^{\mathrm{new}} \end{bmatrix}, \quad b_1^{\mathrm{new}} \in \mathbb{R}^{\Delta h} \tag{5}$$

where $K_1^{(t)}$ serves as an adaptive low-rank project of the input space, and $V_1^{(t)}$ projects it into the newly allocated hidden units $\Delta h$. This allows the model to expand its capacity while keeping previously learned parameters intact, ensuring that new information does not alter existing knowledge. Also, the low-rank projection can effectively capture meaningful information in the input by compressing it into a compact and task-relevant subspace. Similarly, the second-layer weight matrix is expanded to incorporate the additional hidden neurons by:

$$W_2^{(t+1)} = \begin{bmatrix} W_2^{(t)} \\ K_2^{(t)} V_2^{(t)} \end{bmatrix}, \quad K_2^{(t)} \in \mathbb{R}^{\Delta h \times k'}, \quad V_2^{(t)} \in \mathbb{R}^{k' \times h'} \tag{6}$$

where the second-layer bias term remains unchanged as $b_2^{(t+1)} = b_2^{(t)}$, since we preserve the previously learned representation space. Notably, previous parameters $W_1^{(t)}, W_2^{(t)}, b_1^{(t)}, b_2^{(t)}$ remain fixed and are not updated to preserve past knowledge. Only the added parameters $K_1^{(t)}, V_1^{(t)}, K_2^{(t)}, V_2^{(t)}$, and $b_1^{\mathrm{new}}$ are trainable at $t+1$.

This formulation ensures that the MLP's capacity increases gradually without disrupting previously optimized parameters. The forward pass at time $t+1$ now follows:

$$H_{t+1}^{(1)} = \sigma(\mathcal{X}_{t+1} W_1^{(t+1)} + b_1^{(t+1)}) \tag{7}$$

$$H_{t+1}^{(2)} = \sigma(H_{t+1}^{(1)} W_2^{(t+1)} + b_2^{(t+1)}) \tag{8}$$

By incorporating key-value expansion, the model dynamically adapts to new emerging graphs while preserving prior knowledge. The additional parameters are designed to prevent unnecessary redundancy, as the keys and values remain lightweight, introducing minimal computational overhead and ensuring efficient and scalable inference. This progressive expansion approach balances the need for adaptation and inference efficiency. It ensures that the model grows in a controlled and efficient manner, allowing for smooth adaptation without sacrificing computational efficiency.

Apart from the MLPs, each newly expanded MLP at time step $t$ is attached with a prediction head $g_t$, which maps the learned representation to the class space. Specifically, the output representation $H_t^{(2)}$ is passed through $g_t$ to generate class logits as $\hat{\mathcal{Y}}_t = g_t(H_t^{(2)})$. To optimize the corresponding MLPs, we apply the cross-entropy loss as:

$$\mathcal{L}_{\mathrm{MLP}}^t = \mathcal{L}_{\mathrm{CE}}(\hat{\mathcal{Y}}_t, \mathcal{Y}_t), \tag{9}$$

By assigning separate prediction heads to each MLP, the framework enables each MLP to specialize in its corresponding time step and newly emerging node classes.

### 4.3 Energy-based Test-time Routing

In our framework, at each time step $t$, a new MLP model $f_{\mathrm{MLP}}^t$ is constructed by progressively expanding the previous MLP. This results in a series of specialized MLPs, each trained on a distinct subset of the evolving data with a disjoint set of classes. The collection of MLPs naturally forms a *Progressive Mixture of Experts (PMoE)*, where the number of experts dynamically increases over time. For example, at step $t = 0$, a single expert is trained; at step $t = 1$, two experts exist; at step $t = 2$, three experts are available, and so on. Unlike traditional MoE architectures with a fixed set of experts, our design introduces a *growing ensemble*, where new experts are continually added rather than replacing old ones. This PMoE design allows the model to retain previously learned knowledge while efficiently adapting to new graphs. Each expert specializes in the data distribution at its corresponding time step, ensuring that stability and plasticity are balanced without catastrophic forgetting. Compared to single-expert models that continually update a single set of parameters, this formulation provides a *scalable and adaptive learning framework*, where the model increases its capacity as new knowledge is introduced.

Specifically, we propose a test-time routing mechanism by adopting the energy-based model to measure expert confidence directly from the output logits of each MLP. This eliminates the need for canonical gating

training, ensuring that expert selection remains robust to catastrophic forgetting. During test time at time step $t$, given a testing node with feature input $x$, each expert from $\{f_{\mathrm{MLP}}^i\}_{i=0}^t$ produces a prediction by $\hat{y}_i = g_i(f_{\mathrm{MLP}}^i(x))$, where $g_i$ is the prediction head attached to MLP $f_{\mathrm{MLP}}^i$. The confidence of each expert is then quantified using an energy function as:

$$E_i(x) = -\log \sum_c \exp(g_i(f_{\mathrm{MLP}}^i(x))_c) \tag{10}$$

where $g_i(f_{\mathrm{MLP}}^i(x))$ represents the output logits of expert $i$, and the summation is taken over all class logits $c$. Lower energy values indicate higher model confidence, meaning that the expert is more suitable for handling the input node, while higher energy values indicate greater uncertainty in the expert's prediction.

To select the best expert at test time, we identify the expert with the lowest energy score among all experts trained from step 0 to $t$:

$$i^* = \arg \min_{i \in \{0,\ldots,t\}} E_i(x). \tag{11}$$

where $i^*$ denotes the most confident expert up to time step $t$ for the given input node. The final prediction is then defined as $\hat{y}_i = g_{i^*}(f_{\mathrm{MLP}}^{i^*}(x))$. This routing mechanism ensures that only the most confident expert from the available experts at time $t$ is used for inference, preventing interference from less relevant models.

## 4.4 Expert-guided GNN Updating

In our framework, the GNN is updated periodically at a larger time scale compared to the more frequent MLP updates. During each GNN update cycle, the system has already completed multiple MLP update cycles, resulting in a set of progressively expanded MLPs that have adapted to short-term variations in the data. The role of the GNN update is to periodically integrate structural information from the *observed graphs that emerged in the current update cycle* and refine node representations, providing a more robust backbone for future MLP adaptations. Given that GNN training is computationally expensive, it is performed at a lower frequency, ensuring a balance between efficiency and adaptability. However, continual GNN updates inevitably lead to catastrophic forgetting, where new knowledge overwrites previously learned representations, resulting in a loss of past information. This issue arises due to the evolving graph structures, where newly introduced nodes, edges, and labels dynamically shift the model's learned representations, potentially degrading its ability to classify earlier classes.

To address this challenge, we propose leveraging the MLPs built during the current GNN update cycle as knowledge-preserving models to mitigate forgetting in the GNN. Since these experts are trained sequentially, with the first MLP expert preserving past graph data, they inherently retain information from earlier data distributions. Therefore, we use knowledge distillation from the most confident past MLP to guide GNN updating, ensuring that it retains meaningful representations while learning from new data. Specifically, at each GNN update step $T$, we leverage the data accumulated in the current GNN update cycle, denoted as $\mathcal{G}^T = (\mathcal{V}^T, \mathcal{E}^T, X^T, \mathcal{Y}^T)$. The GNN $f_{\mathrm{GNN}}$ learns structure-aware representations by $H_T = f_{\mathrm{GNN}}(\mathcal{G}^T)$. A classification head $g_{\mathrm{GNN}}$ is applied to the learned representations. To ensure the updated GNN retains knowledge from prior tasks, we select the most confident past MLP using the energy-based routing mechanism:

$$i^* = \arg \min_{i \in \{0,\ldots,T\}} E_i(x^T), \tag{12}$$

where $E_i(x^T)$ is the energy score computed for input $x^T$ by the past MLP $f_{\mathrm{MLP}}^i$. The knowledge distillation loss aligns a GNN's prediction with the selected expert's outputs using KL divergence:

$$\mathcal{L}_{\mathrm{GKD}} = \frac{1}{|\mathcal{V}^T|} \sum_{z=1}^{|\mathcal{V}^T|} \mathcal{L}_{\mathrm{KL}}(g_{\mathrm{GNN}}(h_T^z) \parallel g_{i^*}(f_{\mathrm{MLP}}^{i^*}(x_z^T))). \tag{13}$$

The final objective for updating the GNN is a combination of the supervised cross-entropy loss for learning from new labels and the distillation loss for preserving past knowledge:

$$\mathcal{L}_{\mathrm{GNN}} = \mathcal{L}_{\mathrm{CE}} + \beta \mathcal{L}_{\mathrm{GKD}}. \tag{14}$$

Table 2: Overall Performance Comparison. $^*$ denotes $p < 0.05$ for the paired t-test on CMLP with the best baseline.

| Method | CoraFull-CL | | | Arxiv-CL | | | Reddit-CL | | | Products-CL | | |
|---|---|---|---|---|---|---|---|---|---|---|---|---|
| | AP/% ↑ | AF/% ↑ | H-Mean ↑ | AP/% ↑ | AF/% ↑ | H-Mean ↑ | AP/% ↑ | AF/% ↑ | H-Mean ↑ | AP/% ↑ | AF/% ↑ | H-Mean ↑ |
| MLP | 12.83 | -77.00 | 16.47 | 12.56 | -68.74 | 17.91 | 15.96 | -80.34 | 17.61 | 15.74 | -78.63 | 18.12 |
| GraphMLP | 12.89 | -77.63 | 16.35 | 13.25 | -68.36 | 18.67 | 18.17 | -83.87 | 17.08 | 16.20 | -79.13 | 18.24 |
| GLNN | 13.24 | -76.71 | 16.88 | 13.70 | -69.08 | 18.98 | 18.48 | -84.03 | 17.13 | 16.27 | -79.47 | 18.15 |
| GENN | 13.85 | -77.89 | 17.03 | 14.31 | -69.59 | 19.46 | 19.20 | -84.72 | 17.02 | 16.58 | -80.40 | 17.96 |
| VQGraph | 14.05 | -78.61 | 16.95 | 14.85 | -69.94 | 19.87 | 19.58 | -84.58 | 17.25 | 16.87 | -81.28 | 17.74 |
| NOSMOG | 14.18 | -78.59 | 17.06 | 14.63 | -68.86 | 19.90 | 19.38 | -84.49 | 17.23 | 16.62 | -80.81 | 17.81 |
| ERGNN | 20.18 | -65.74 | 25.40 | 17.28 | -46.30 | 26.15 | 30.69 | -42.51 | 40.02 | 21.37 | -42.56 | 31.15 |
| GEM | 17.68 | -71.55 | 21.80 | 15.63 | -65.84 | 21.44 | 25.60 | -60.67 | 31.01 | 18.69 | -51.46 | 26.98 |
| MAS | 20.79 | -68.84 | 24.93 | 15.42 | -64.83 | 21.43 | 27.18 | -52.40 | 34.60 | 19.05 | -50.83 | 27.46 |
| LwF | 14.89 | -75.15 | 18.62 | 15.25 | -65.80 | 21.09 | 23.58 | -68.37 | 27.01 | 18.63 | -52.77 | 26.72 |
| EWC | 23.64 | -65.56 | 28.03 | 17.32 | -42.56 | 26.61 | 32.50 | -45.89 | 40.60 | 20.21 | -45.50 | 29.48 |
| **CMLP** | **42.66**$^*$ | **-14.54**$^*$ | **56.91**$^*$ | **23.48**$^*$ | **-9.53**$^*$ | **37.28**$^*$ | **55.84**$^*$ | **-12.80**$^*$ | **66.73**$^*$ | **25.59**$^*$ | **-22.71**$^*$ | **38.44**$^*$ |

where $\beta$ controls the balance between learning new information and retaining prior knowledge. By periodically updating the GNN at a larger time scale, the model captures long-term structural dependencies in the evolving graph while leveraging the continuously updated MLPs for efficient inference in between GNN updates. The asynchronous updates ensure that both short-term adaptability and long-term structural learning are balanced in an efficient manner while preventing catastrophic forgetting in the framework. Note that at the end of each GNN update cycle, all expanded MLPs are discarded to reduce memory usage, and a new MLP is reinitialized from scratch for Graph-to-MLP distillation to begin the next MLP cycle.

# 5 Experiments

## 5.1 Experimental Setup

**Dataset.** We evaluate our proposed method on four benchmark datasets from CGLB (Zhang et al., 2022c). CoraFull-CL (Bojchevski & Günnemann, 2018) consists of 70 classes, Arxiv-CL (Hu et al., 2020) and Reddit-CL (Hamilton et al., 2017) each contain 40 classes, and Products-CL (Hu et al., 2020) includes 46 classes. Each dataset is partitioned into five groups: the first is used to train the GNN and perform GNN-to-MLP distillation, while the remaining groups are incrementally introduced to update the MLPs in subsequent steps. Additional dataset details are provided in Appendix D.

**Baselines.** To evaluate **CMLP**, we compare it with state-of-the-art models across two main categories, including *GNN-to-MLP distillation* and *continual learning*. The first category includes **GLNN** (Zhang et al., 2022b), **GraphMLP** (Hu et al., 2021), **GENN** (Wang et al., 2023), **VQGraph** (Yang et al., 2024), **NOSMOG** (Tian et al., 2022), and **ERGNN** (Zhou & Cao, 2021). The second category consists of **GEM** (Lopez-Paz & Ranzato, 2017), **MAS** (Aljundi et al., 2018), **LwF** (Li & Hoiem, 2017), and **EWC** (Kirkpatrick et al., 2017). All these models are evaluated under identical experimental setups with the same evolving graphs and update schedule as **CMLP**, to ensure a fair comparison. Additional details about baselines are provided in Appendix D.

**Experimental Details.** We follow an evaluation protocol tailored for graph continual learning (Zhang et al., 2022c), and employ *Average Performance (AP), Average Forgetting (AF)*, and *H-mean* as evaluation metrics. Due to the computational overhead of GNNs, GNN updates occur less frequently (e.g., every 5 days), while MLP is updated more frequently (e.g., daily) to rapidly adapt to new data and provide fast, timely inference. To balance efficiency and adaptability, we restrict the ratio of $C^{\mathrm{G}}/C$ to remain below 10. This setup ensures that MLP quickly captures short-term variations, while the GNN periodically integrates structural information at a larger time scale. In this work, we focus on the incremental learning process of MLP within each GNN update cycle, ensuring that the inference model rapidly adapts to evolving data distributions while leveraging previously acquired knowledge. Details about the evaluation protocol and implementation are provided in the Appendix D.

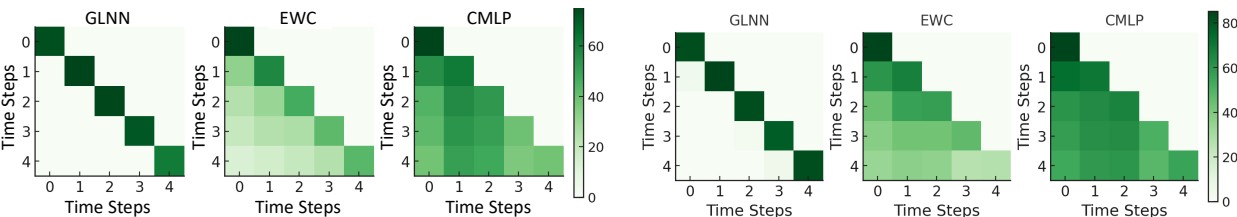

Figure 3: Performance heatmaps over time on Cora (left) and Reddit (right). Each entry $M_{i,j}$ denotes the classification accuracy on classes introduced at time step $j$, evaluated after training up to time step $i$. The diagonal entries ($i = j$) reflect accuracy on newly learned classes, while the lower-triangular region ($i > j$) reflects accuracy on previously learned classes (knowledge retention). Stronger lower-triangular values indicate better retention of past knowledge.

## 5.2 Experimental Results

**Overall Performance.** To assess the effectiveness of our proposed framework, we compare CMLP against both GNN-to-MLP baselines and continual learning baselines in their ability to continuously learn and adapt MLPs. As shown in Table 2, *CMLP significantly outperforms all baseline models*. While existing GNN-to-MLP methods incorporate different techniques to embed graph structure into MLPs, their poor performance on the class-incremental graph highlights their inability to handle the continual adaptation of MLPs to evolving graphs, leading to high forgetting scores. On the other hand, continual learning baselines perform better than GNN-to-MLP models, as they adopt various strategies to mitigate catastrophic forgetting. However, their reliance on selecting representative samples or estimating parameter importance still inevitably modifies previously learned parameters, ultimately degrading continual learning performance. In contrast, CMLP preserves all learned parameters intact and adopts a progressive MLP expansion strategy, ensuring the retention of past knowledge while adapting to newly emerging node classes. This design makes CMLP particularly well-suited for the continual learning of MLPs in the class-incremental setting.

**Performance Over Time**. We present the performance matrix $M$ comparisons between GLNN, EWC, and CMLP in Figure 3 to analyze how classification accuracy evolves over time. Each entry $M_{i,j}$ in the heatmap represents the classification accuracy of a model on the novel node classes introduced at time step $j$, after training on the emerging graph at time step $i$. In particular, the diagonal entries ($i = j$) measure performance on newly introduced classes, while the lower-triangular entries ($i > j$) measure performance on previously learned classes and thus reflect knowledge retention. To correctly interpret the heatmaps, it is important to note that darker colors on the diagonal indicate strong performance on the current task, whereas the lower-triangular region captures the extent of forgetting. A model with stronger lower-triangular values retains more past knowledge over time. From these comparisons, we observe that as time progresses, *CMLP demonstrates a stronger ability to retain past knowledge and achieves superior classification accuracy on previously learned classes* compared to GLNN and EWC. GLNN primarily focuses on learning from the current input graphs while entirely forgetting past knowledge.

EWC, although more effective at mitigating forgetting than GLNN, suffers from inaccuracies in estimating parameter importance, limiting its ability to fully preserve prior knowledge. In contrast, CMLP effectively maintains past class information while continually adapting to new data, leading to more stable long-term performance.

**Performance of GNN**. Since our experiments include five MLP update cycles within one GNN update cycle, we evaluate the GNN's performance on the classes that appeared in the stage $t_0$, where GNN-to-MLP distillation was performed. As Figure 4 shows, *GNN at $t_0$* represents the classification accuracy of the initial GNN model used for distillation. *GNN w/o Expert* refers to the updated

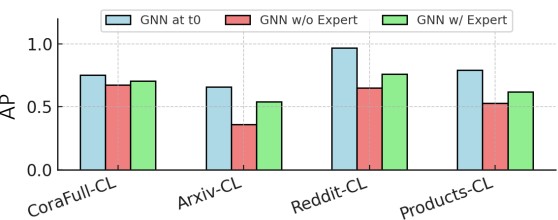

Figure 4: GNN performance across datasets, where "w/ Expert" shows how MLP guidance mitigates teacher forgetting across update cycle.

Table 3: The results of ablation studies.

| Method | CoraFull-CL | | | Arxiv-CL | | |
|---|---|---|---|---|---|---|
| | AP/% ↑ | AF/% ↑ | H-Mean ↑ | AP/% ↑ | AF/% ↑ | H-Mean ↑ |
| Single MLP | $12.83 \pm 0.65$ | $-77.00 \pm 2.43$ | 16.47 | $12.56 \pm 0.37$ | $-68.74 \pm 2.38$ | 17.91 |
| CMLP w/o E | $18.54 \pm 0.84$ | $-68.56 \pm 1.57$ | 23.33 | $16.30 \pm 0.48$ | $-57.55 \pm 1.92$ | 23.56 |
| CMLP | $42.66 \pm 1.32$ | $-14.54 \pm 0.47$ | 56.91 | $23.48 \pm 0.61$ | $-9.53 \pm 0.38$ | 37.28 |

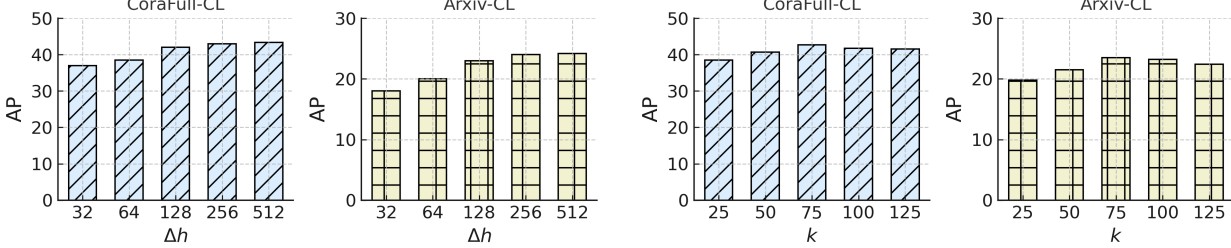

Figure 5: The impact of expansion dimension $\Delta h$.      Figure 6: The impact of low-rank projection $k$.

GNN model after MLP update cycles, but without guidance from the trained MLPs. *GNN w/ Expert* represents the updated GNN model that incorporates knowledge from PMOE during its update. From the results, we observe that the GNN undergoes catastrophic forgetting after completing a GNN update cycle, as indicated by the performance drop when comparing GNN at $t_0$ with the other two variants. This forgetting occurs because the parameters of the GNN are modified during the update. However, with guidance from the MLPs trained throughout the cycle, the forgetting effect is mitigated. The mixture of experts effectively preserves past knowledge acquired at step $t_0$ and facilitates knowledge distillation back into the GNN, helping it retain previously learned information.

**Ablation study.** To assess the contribution of each component in CMLP, we conduct an ablation study by comparing CMLP with two variants. The first variant, **Single MLP**, maintains only a single MLP that is continually updated as new graphs emerge, without any explicit mechanism for preserving past knowledge. The second variant, **CMLP w/o E**, removes the mixture of experts design, relying solely on the latest MLP at each time step for inference. As shown in Table 3, **Single MLP** performs the worst, as continual updates overwrite previously learned knowledge, leading to severe forgetting. **CMLP w/o E** achieves slightly improvement over Single MLP but remains inferior to the full CMLP, as the latest MLP primarily adapts to the current graph while disregarding prior knowledge. These results show the importance of the mixture of experts in preserving historical information and ensuring long-term performance.

**Impact of MLP Expansion Parameters.** To investigate the impact of the MLP expansion dimension $\Delta h$, we conduct experiments with different expansion sizes ranging from 32 to 512 on CoraFull-CL and Arxiv-CL. As shown in Figure 5, we observe that performance improves as the expansion dimension increases. When the expansion dimension is small (e.g., 32), the model has a limited set of parameters for learning from emerging graphs and classes, leading to insufficient network capacity and poor adaptation to new data. In contrast, when the expansion dimension is larger (e.g., 256 or 512), the performance significantly improves compared to the 32-dimensional setting, indicating that a larger network provides better learning capacity. Additionally, we find that the performance with 256 and 512 dimensions is similar, suggesting that a 256-dimensional expansion is sufficient for learning new data under the current experimental setting. Furthermore, we also study the impact of the MLP low-rank projection parameter $k$. Similarly, we progressively expand the MLP with varying values of $k \in \{25, 50, 75, 100, 125\}$, which control the parameter in the key-value expansion. As shown in Figure 6, the performance generally improves with increasing $k$, especially when moving from 25 to 75, indicating that higher-rank projections allow for richer capacity and more expressive updates. However, further increasing $k$ beyond 75 leads to marginal or even slightly decreased improvements, suggesting a saturation point where the additional projection dimensions may not yield significant gains and

Table 4: The impact of the number of MLP update cycles.

| Update Cycles | CoraFull-CL | | | Reddit-CL | | |
|---|---|---|---|---|---|---|
| | AP/% ↑ | AF/% ↑ | H-Mean↑ | AP/% ↑ | AF/% ↑ | H-Mean↑ |
| 3 | $47.52 \pm 1.49$ | $-10.70 \pm 0.42$ | 62.03 | $58.41 \pm 1.47$ | $-11.01 \pm 0.58$ | 70.53 |
| 5 | $42.66 \pm 1.32$ | $-14.54 \pm 0.47$ | 56.91 | $55.84 \pm 1.02$ | $-12.80 \pm 0.52$ | 66.73 |
| 7 | $36.45 \pm 1.37$ | $-14.84 \pm 0.59$ | 51.05 | $52.38 \pm 1.38$ | $-14.27 \pm 0.60$ | 65.03 |

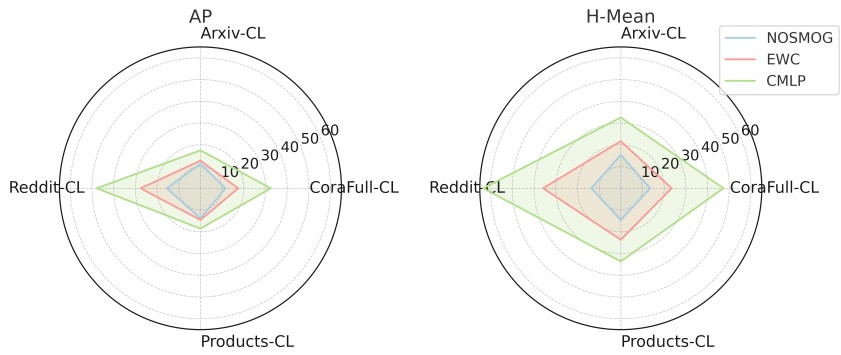

Figure 7: Performance comparison (AP and H-Mean) on four datasets when the MLP update cycle is fixed at 10.

could introduce redundancy. This trend is consistently observed across both the CoraFull-CL and Arxiv-CL, demonstrating that moderate values of $k$ strike a favorable balance between expressiveness and efficiency.

**Impact of the Number of MLP Update Cycles.** To explore the influence of MLP update cycles, we conduct experiments on CoraFull-CL and Reddit-CL with varying update cycles of 3, 5, and 7. As shown in Table 4, we observe that the AP score decreases as the update cycle increases, indicating a decline in adaptation ability. Additionally, the average forgetting increases since the energy-based confidence estimation becomes less distinguishable when the number of experts grows. These results suggest that a smaller update cycle leads to better overall performance but requires more frequent GNN updates. Conversely, a larger update cycle degrades performance due to accumulated forgetting. Therefore, it is crucial to balance the GNN update cycle and the MLP update cycle to achieve an optimal tradeoff between adaptation and knowledge retention. In practice, short-to-moderate MLP update intervals define the most stable operating region, offering a good balance between rapid adaptation, routing stability, and knowledge retention. Moreover, we also compare CMLP with the two best baseline models in two categories, including NOSMOG and EWC, under the setting where MLP update cycle is fixed at 10. As illustrated in Figure 7, CMLP consistently outperforms both baselines across all four benchmarks on both AP and H-Mean metrics. These results highlight that CMLP is more robust under larger MLP update cycles, where baseline methods tend to suffer from more severe forgetting due to their limited mechanisms for preserving past knowledge. In contrast, CMLP benefits from its progressive MLP expansion, which mitigates forgetting while ensuring continuous adaptation.

## 6 Complexity Analysis

**Time Complexity Analysis** To support continual adaptation on class-incremental graphs while maintaining inference efficiency, we adopt a key-value expansion strategy that incrementally increases the model's capacity without modifying previously learned parameters. During the training phase at each time step $t$, only the newly introduced key-value matrices $K_1^{(t)}, V_1^{(t)}$ and $K_2^{(t)}, V_2^{(t)}$ are updated, while the previously learned weights $W_1^{(t)}, W_2^{(t)}, b_1^{(t)}, b_2^{(t)}$ remain fixed to preserve past knowledge. The computational complexity

of training primarily involves computing the key-value products $K_1^{(t)} V_1^{(t)} \in \mathbb{R}^{d \times \Delta h}$ and $K_2^{(t)} V_2^{(t)} \in \mathbb{R}^{\Delta h \times h'}$, in addition to the forward computations through the expanded layers. This results in an overall training time complexity defined as:

$$O(md(h + \Delta h) + m(h + \Delta h)h' + dk + k\Delta h + \Delta h k' + k'h'), \tag{15}$$

where $m$ is the batch size, $d$ the input dimension, $h$ the base hidden size, $h'$ the output size, and $k, k'$ the low-rank projection dimensions. Since $k$ and $k'$ are typically much smaller than $d$, $h$, and $h'$, the added cost remains modest and scales linearly with the expansion size $\Delta h$. During inference, all previously learned and newly added parameters are used. If the key-value products are precomputed and cached after training, the inference time complexity reduces to a standard forward pass with cost as:

$$O(md(h + \Delta h) + m(h + \Delta h)h'). \tag{16}$$

This formulation enables progressive model growth with limited overhead, allowing efficient and scalable inference. In addition, routing requires evaluating all available experts and computing one energy score per expert. Therefore, the total compute grows approximately linearly with the number of experts. For example, the inference time of the last expert is approximately $5\times$ that of the first expert when there are five MLP update cycles. However, since energy computation itself is only a lightweight log-sum-exp reduction over logits, the dominant cost comes from the expert forward passes. In practice, these expert evaluations can be parallelized, so the latency is mainly determined by the largest expert and a small reduction step for selecting the minimum-energy expert. Nevertheless, because the expert remains lightweight, the absolute inference time of the last expert is still expected to stay below 1 ms on modern GPUs, so the practical increase is negligible.

**Parameter Complexity Analysis** We analyze the number of parameters introduced by the key-value expansion strategy to evaluate its efficiency and scalability. At each time step $t$, the MLP model has a first-layer weight matrix $W_1^{(t)} \in \mathbb{R}^{d \times h}$, a second-layer weight matrix $W_2^{(t)} \in \mathbb{R}^{h \times h'}$, and associated bias terms $b_1^{(t)} \in \mathbb{R}^h$ and $b_2^{(t)} \in \mathbb{R}^{h'}$. The total number of parameters before expansion is defined as:

$$P_{\text{base}} = d \cdot h + h + h \cdot h' + h'. \tag{17}$$

At time step $t + 1$, the model is expanded by adding $\Delta h$ hidden units. Instead of directly appending new weight blocks $W_{\text{new}} \in \mathbb{R}^{d \times \Delta h}$, we adopt a key-value decomposition strategy: introducing a key matrix $K_1^{(t)} \in \mathbb{R}^{d \times k}$ and a value matrix $V_1^{(t)} \in \mathbb{R}^{k \times \Delta h}$, along with a bias $b_1^{\text{new}} \in \mathbb{R}^{\Delta h}$. Similarly, the second layer is expanded using $K_2^{(t)} \in \mathbb{R}^{\Delta h \times k'}$ and $V_2^{(t)} \in \mathbb{R}^{k' \times h'}$. The total number of new parameters introduced is:

$$P_{\text{added}} = d \cdot k + k \cdot \Delta h + \Delta h + \Delta h \cdot k' + k' \cdot h'. \tag{18}$$

Compared to a naive expansion method that directly introduces $W_{\text{new}} \in \mathbb{R}^{d \times \Delta h}$ with cost $O(d \cdot \Delta h)$, our key-value approach achieves reduced parameter growth of $O(d \cdot k + k \cdot \Delta h)$, assuming $k \ll \Delta h$. This structured decomposition reduces redundancy and provides an efficient and compact alternative for continual model expansion.

## 7 Conclusion

In this paper, we proposed CMLP, a novel framework for GNN-to-MLP distillation in class-incremental graph learning. To achieve fast adaptation and long-term knowledge retention, CMLP introduces an asynchronous update paradigm that decouples GNN and MLP training. We further developed a progressive MLP expansion strategy, allowing the MLP to adapt to new classes while preserving low-latency inference. An energy-based test-time routing mechanism was designed for efficient expert selection. Additionally, to alleviate GNN forgetting, the trained MLP experts are used as teachers for distillation back into the GNN. Experiments on real-world datasets show the effectiveness of CMLP on bridging inference efficiency and adaptability.

## Acknowledgments

This work was supported by the National Science Foundation (award number 2451605). Any opinions, findings, conclusions, or recommendations expressed in this material are those of the authors and do not necessarily reflect the views of the NSF.

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

## A    More Detailed Related Work

In this section, we survey the related studies on GNN-to-MLP distillation and graph class-incremental learning.

## A.1 GNN-to-MLP Distillation

GNN-to-MLP distillation (Zhang et al., 2022b; Wang et al., 2023) has emerged as a promising approach to enable graph-free inference by training a lightweight MLP to mimic the predictions of a GNN while preserving structural knowledge. This paradigm is particularly appealing in scenarios where low-latency inference is critical, as the recursive message passing operations in GNNs are computationally expensive and often impractical for real-time applications or resource-constrained environments. To address this challenge, GLNN (Zhang et al., 2022b) proposed distilling the output of a trained GNN into a structure-independent MLP, effectively removing the need for message passing at inference time. While this results in significant speedups, it also eliminates the inference model's direct access to relational information, thereby limiting its ability to fully capture the structural patterns present in graph data. To compensate for the loss of explicit structural cues, a number of follow-up studies have explored mechanisms for injecting structural inductive biases into the MLP during distillation. GENN and NOSMOG (Wang et al., 2023; Tian et al., 2022) incorporate positional embeddings derived from graph Laplacians or random walks to encode node location and topological context. CPF (Yang et al., 2021) applies label propagation during training to approximate local smoothness and improve decision boundaries. A separate line of work focuses on neighborhood alignment, where the goal is to align the representations or outputs of MLPs with those of their GNN teachers at the local subgraph level. Representative methods include (Xiao et al.; Dong et al., 2022; Hu et al., 2021; Liu et al., 2022), which apply contrastive learning, output matching, or feature consistency constraints to preserve local relational signals. More recently, structural discretization approaches have been proposed. VQGraph (Yang et al., 2024) introduces a vector-quantized codebook that learns to represent common subgraph motifs, using these discrete codes to guide MLP training and encode structural priors. This method achieves strong performance while maintaining scalability. In parallel, AdaGMLP (Lu et al., 2024) introduces an ensemble-based framework for distillation, where multiple MLPs are trained on random node subsets and combined using an AdaBoost-inspired strategy. AdaGMLP further incorporates a node alignment mechanism to ensure hidden and output-level consistency, particularly in the presence of sparse or incomplete node features. Recent work, such as RbM (Rumiantsev & Coates), also explores MoE-style student architectures to enhance graph-free inference. Our PMoE differs significantly in motivation and execution. RbM focuses on a static setting, jointly training experts within a dedicated MoE student to improve performance on fixed graphs via memory-based routing. In contrast, PMoE is designed for the class-incremental setting; our experts are not trained from scratch but emerge from a progressive chain of expanded MLPs over time. Both methods share the high-level intuition of selecting the most suitable expert at test-time, yet PMoE specifically leverages this mechanism to balance new class adaptation with prior knowledge retention, a challenge outside the scope of static MoE distillation.

Despite these advances, existing GNN-to-MLP distillation methods are largely confined to static graphs, where distillation is performed once from a pre-trained GNN and the MLP does not evolve further. In contrast, real-world graphs are dynamic in nature, with new nodes and classes emerging continuously. These evolving settings demand models that can not only perform fast inference but also continually adapt without sacrificing previously acquired knowledge. However, current approaches overlook the challenge of supporting efficient inference and incremental knowledge retention in the class-incremental graph setting.

## A.2 Graph Class-incremental Learning

Graph class-incremental learning (GCIL) extends traditional incremental learning (De Lange et al., 2021; Goswami et al., 2024; Wang et al., 2024; Chen et al., 2024) to dynamic graph settings, where models must continually learn from new node classes and evolving graph structures while preserving performance on previously seen data. A key challenge in GCIL is catastrophic forgetting, where updates on new tasks can overwrite prior knowledge. Existing methods largely follow three canonical approaches. (1) Regularization-based methods (Li & Hoiem, 2017; Kirkpatrick et al., 2017; Liu et al., 2021) impose constraints on important model parameters to reduce forgetting, such as penalizing changes to parameters critical to earlier tasks or maintaining structural consistency in learned embeddings. (2) Memory-based methods explicitly store a portion of previous data to replay during training; for example, ERGNN (Zhou & Cao, 2021) selects representative nodes using coverage and influence maximization strategies to build a replay buffer, SSM (Zhang et al., 2022e) maintains sparsified subgraphs to capture essential topology with minimal memory, and CaT

(Liu et al., 2023) constructs compact synthetic graphs via condensation and then trains exclusively on these graphs, effectively addressing class imbalance during replay. (3) Architectural methods dynamically expand the model to accommodate new tasks (Zhang et al., 2024a), as in the hierarchical prototype-based framework proposed in (Zhang et al., 2022d), where parameter subsets are selectively activated based on task identity. Beyond these categories, recent work has proposed alternative strategies to further improve GCIL performance. Replay-and-Forget-Free GCIL (Niu et al., 2024) introduces a task profiling and prompting mechanism that enables class-incremental learning without explicit replay, using task-specific prompts to encode knowledge and condition predictions. Meanwhile, GSIP (Li et al., 2024) provides an information-theoretic perspective, arguing that preserving discriminative and structural signals is more effective than naive example replay, and proposing a benchmark and training strategy to explicitly enforce this principle. And GCAL (Qiao et al.) studies continual adaptation under evolving graph domain shifts and uses bilevel optimization together with generated graph memories to retain prior-domain knowledge during sequential adaptation. Despite these advances, existing graph class-incremental learning methods have primarily focused on updating a single GNN model with sufficient capacity to retain past knowledge, which differs significantly from our problem setting. Applying GNN-to-MLP distillation on class-incremental graph remains unexplored. Furthermore, existing methods often rely on complex mechanisms for selecting representative data or identifying important parameters to preserve prior knowledge. However, these selection processes are inherently imperfect, and errors in data or parameter selection can compromise the model's ability to retain previously acquired information, leading to degraded performance over time.

## B   Additional Preliminary

**Definition 4 (GNN-to-MLP Knowledge Distillation)** *GNN-to-MLP knowledge distillation transfers knowledge from a GNN (teacher) to a lightweight Multi-Layer Perceptron (MLP) (student) by minimizing the discrepancy between their output logits, enabling graph-free inference while retaining essential structural information. The student model significantly reduces inference time and resource consumption, making it well-suited for large-scale applications, low-latency services, and environments where graph connectivity is unavailable.*

## C   Optimization

We first distill knowledge from a trained GNN into an MLP by optimizing Eq.(1), enabling graph-free inference. As new graph data arrives, the MLP progressively expands following Eq.(4)-(6), allowing it to adapt to new classes while preserving previously learned parameters. The GNN is then periodically updated using newly accumulated graph data, with past MLPs providing guidance to mitigate catastrophic forgetting, as formulated in Eq.(14). During inference, the energy-based routing mechanism selects the most confident MLP as the expert, ensuring efficient expert utilization without relying on labeled data. The training procedure of CMLP is provided in Algorithm 1.

## D   Additional Experiment Setup

### D.1   Datasets

We evaluate our proposed method on four benchmark datasets from CGLB (Zhang et al., 2022c). The first dataset, CoraFull-CL (Bojchevski & Günnemann, 2018), consists of 70 classes, which we divide into five groups of 14 new classes per cycle, corresponding to five MLP update cycles within one GNN update cycle. Specifically, the first 14 classes are used for training the GNN and performing GNN-to-MLP distillation, while the remaining 56 classes are introduced incrementally to update the MLPs in subsequent steps. The second and third datasets, Arxiv-CL (Hu et al., 2020) and Reddit-CL (Hamilton et al., 2017), each contain 40 classes, which are similarly divided into five groups of eight new classes per cycle. In each case, the first eight classes are used to train the GNN and distill knowledge into the MLP, while the remaining classes are incrementally introduced to update the MLP in later steps. The last dataset, Products-CL (Hu et al., 2020),

---

**Algorithm 1:** Training Procedure of CMLP

---

**Input:** An evolving dataset $\{\mathcal{G}_t\}_{t=0}^T$, $C^{\mathrm{G}} = T$ and $C = 1$, GNN teacher $f_{\mathrm{GNN}}(\cdot)$, MLP student $f_{\mathrm{MLP}}^0(\cdot)$

/* GNN-to-MLP Distillation                                            */

Train $f_{\mathrm{GNN}}(\cdot)$ and $f_{\mathrm{MLP}}^0(\cdot)$ on $\mathcal{G}_0$ and optimize loss function $\mathcal{L}_{\mathrm{KD}}$ defined in Eq. (1);;

**foreach** *time step* $t \in \{1, 2, \ldots, T\}$ **do**

     /* Progressive MLP Expansion                                   */

     Conduct the progressive MLP expansion following Eq. (4)-(6) and obtain a new MLP $f_{\mathrm{MLP}}^t(\cdot)$;;

     Train the new MLP $f_{\mathrm{MLP}}^t(\cdot)$ on $\mathcal{G}_t$ using Eq. (9);;

     /* Energy-based Test-time Routing                           */

     Collect $\{f_{\mathrm{MLP}}^i\}_{i=0}^t$ to form PMOE;;

     Calculate the confidence of each expert using the energy function in Eq. (10);;

     Select the best expert MLP for inference following Eq. (11);;

/* Expert-guided GNN Updating                                      */

Optimize GNN teacher $f_{\mathrm{GNN}}(\cdot)$ on accumulated graph $\mathcal{G}^T$ using Eq. (12)-(14);

**return** GNN $f_{\mathrm{GNN}}(\cdot)$ for the next update cycle;

---

Table 5: The statistics of the benchmark datasets

| Datasets | CoraFull-CL | Arxiv-CL | Reddit-CL | Products-CL |
|---|---|---|---|---|
| # nodes | 19,793 | 169,343 | 227,853 | 2,449,028 |
| # edges | 130,622 | 1,166,243 | 114,615,892 | 61,859,036 |
| # classes | 70 | 40 | 40 | 46 |

includes 46 classes. Following the same pipeline, we divide these classes into five groups corresponding to five update cycles, with one group containing 10 classes for training the GNN and conducting knowledge distillation, and the remaining four groups each containing 9 classes for later MLP updates. The statistics of the benchmark datasets are provided in Table 5.

### D.2 Evaluation Protocol

We follow an evaluation protocol tailored for continual learning (Zhang et al., 2022c), where models are initially trained on the first graph $\mathcal{G}_0$ and subsequently updated with newly emerging node classes from $\mathcal{G}_1, \mathcal{G}_2, \ldots, \mathcal{G}_T$. At each time step $t$, the model is updated using the current graph $\mathcal{G}_t$, while access to previous graphs $\mathcal{G}_0, \ldots, \mathcal{G}_{t-1}$ is restricted, simulating a realistic class-incremental setting where historical data is unavailable. To evaluate the model's ability to balance adaptability and knowledge retention, we construct a performance matrix $M \in \mathbb{R}^{T \times T}$, where each entry $M_{i,j}$ represents the classification accuracy on node classes introduced at time step $j$ after training on $\mathcal{G}_i$. Based on this, we employ two evaluation metrics (Zhang et al., 2022c). The *Average Performance (AP)* is given by:

$$\mathrm{AP}_i = \frac{1}{i} \sum_{j=1}^{i} M_{i,j}, \quad i = 1, \ldots, T, \tag{19}$$

which measures the model's average accuracy across all encountered node classes after training on graph $\mathcal{G}_i$, assessing its overall learning capability. The *Average Forgetting (AF)* is computed as:

$$\mathrm{AF}_i = \frac{1}{i-1} \sum_{j=1}^{i-1} (M_{i,j} - M_{j,j}), \quad i = 2, \ldots, T, \tag{20}$$

which quantifies the degree of forgetting for previously learned node classes, computed as the drop in accuracy from the time a node class was first learned to the current evaluation step. The AP and AF after learning all T tasks are reported. To provide a single numerical measure of overall model performance, we compute

the *H-mean*, which takes the harmonic mean of *AP* and *AF*. Given that *AF* is negative, we adjust it by adding 100 before computing the *H-mean*, ensuring a meaningful and interpretable evaluation metric.

### D.3 Baselines

To comprehensively evaluate our proposed CMLP, we compare it with state-of-the-art models across two main categories, including *GNN-to-MLP distillation methods* and *continual learning strategies*. The first category includes *GNN-to-MLP distillation models*, which aim to enable efficient, graph-free inference by transferring knowledge from GNNs to MLPs:

- **GraphMLP** (Hu et al., 2021) injects structural information into MLPs using node-level feature propagation and positional encodings, enabling effective node classification without relying on message passing.

- **GLNN** (Zhang et al., 2022b) distills a pre-trained GNN into a structure independent MLP, accelerating inference while preserving performance by eliminating the need for explicit graph structure during inference.

- **GENN** (Wang et al., 2023) improves distillation by incorporating Laplacian eigenvector-based positional embeddings into the MLP, aiming to retain global structural context from the original graph.

- **VQGraph** (Yang et al., 2024) employs vector quantization to encode discrete motif-based patterns, guiding MLP training with quantized structural features that capture local subgraph semantics.

- **NOSMOG** (Tian et al., 2022) enhances the robustness of the distillation process by applying noise-injected message passing, generating more stable supervision signals for MLPs without structural input.

- **ERGNN** (Liu et al., 2021) mitigates catastrophic forgetting by selecting representative nodes from previous tasks and replaying them during training to preserve knowledge of earlier classes.

All these models are evaluated under identical experimental setups with the same evolving graphs and update schedule as CMLP, to ensure a fair comparison. Specifically, the direct continual adoption of standard GNN-to-MLP distillation requires repeatedly fine-tuning the teacher GNN and re-distilling a new MLP whenever new graph data arrive. To adapt ERGNN to our setting, we follow its experience replay strategy by selecting representative nodes from previous steps and storing them in a replay buffer. During each update, the teacher GNN is trained on both the new nodes and the replayed nodes, after which the updated teacher is distilled into the student MLP for graph-free inference.

The second category consists of *continual learning methods*, which are applied to a single MLP to examine their effectiveness in mitigating catastrophic forgetting:

- **GEM** (Lopez-Paz & Ranzato, 2017) stores a small memory of past examples and constrains gradients during training to preserve previously acquired knowledge.

- **MAS** (Aljundi et al., 2018) estimates parameter importance and applies weight-based regularization to discourage updates to critical parameters.

- **LwF** (Li & Hoiem, 2017) distills knowledge from previous tasks using soft targets, avoiding the need to store past data explicitly.

- **EWC** (Kirkpatrick et al., 2017) remembers old tasks by selectively slowing down learning on the weights important for those tasks.

These continual learning baselines provide comparisons for understanding how well CMLP maintains performance over time compared to traditional forgetting-mitigation approaches. The continual learning baselines are selected according to the target model studied in this work. Since our setting focuses on continually

adapting a distilled MLP student for graph-free inference, rather than continually updating a single GNN, we use architecture-agnostic methods such as GEM, MAS, LwF, and EWC as the main baselines. These methods can be applied directly to the evolving student MLP under the same update schedule as our approach, providing controlled references for the student-learning component. By contrast, existing graph class-incremental methods typically operate on graph models themselves and rely on graph-specific mechanisms, making them less directly comparable to our target setting.

### D.4 Implementation Details

All experiments were conducted on a workstation equipped with Nvidia GeForce RTX 4090 GPUs. We follow the dataset preprocessing and splitting pipeline provided in CGLB (Zhang et al., 2022c). For the models, we use a basic GCN with a 256-dimensional embedding and a two-layer MLP for distillation. In the progressive MLP expansion, we set $\Delta h = 256$ and low-rank project $k = 75$. The Adam optimizer is employed with a learning rate of 0.001. For the tradeoff parameters $\alpha$ and $\beta$, we performed a search over $\{0.01, 0.1, 1, 10\}$ and found that setting both parameters to 1 achieved the best performance. To ensure robustness, all experiments were repeated three times, and we report the average results. We also determine the best parameter configurations for all baseline methods to ensure a fair comparison.

## E Additional Experimental Results

### E.1 Impact of the Different Teacher GNNs

To evaluate the influence of the teacher GNN architectures on the performance of the MLP, we compare four commonly used GNN models: GCN (Kipf & Welling, 2017), GraphSAGE (Hamilton et al., 2017), GAT (Veličković et al., 2018b), and APPNP (Gasteiger et al., 2018). These models are evaluated across four datasets used in the paper. As illustrated in Figure 8, the performance differences among the four GNN variants are relatively minor across all datasets. This suggests that the choice of teacher model has only a limited impact on the final performance of the MLP. This observation can be attributed to the design of our framework, where the GNN-to-MLP distillation is conducted only once at the beginning of each GNN update cycle. After

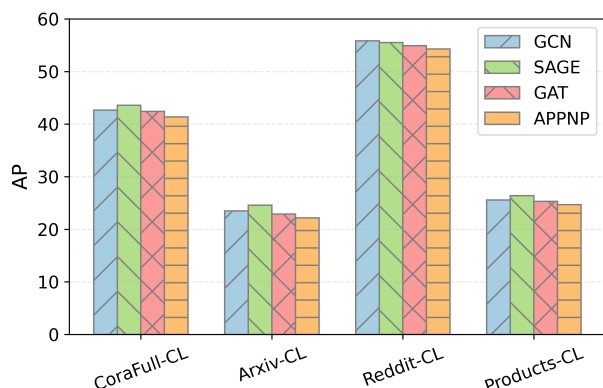

Figure 8: The impact of different teacher GNNs.

distillation, the MLP evolves independently through the proposed progressive expansion strategy, which enables it to incrementally adapt to newly emerging classes. Consequently, the performance of the MLP is primarily determined by its own adaptive updates rather than by the specific GNN used for initial supervision. This finding highlights the robustness of our method and its ability to operate effectively with a variety of GNN backbones.

### E.2 Further Ablation Studies

To better isolate the contribution of the proposed MLP expansion mechanism, we further consider two additional ablations. First, we evaluate a variant that retains progressive expansion but removes the low-rank decomposition, so that each expansion is implemented with standard full-parameter updates. Second, we evaluate a non-expanding MLP whose size is fixed from the beginning to match the final size obtained after expansions in our full model. This ablation controls for total model capacity and tests whether the gain comes merely from having a larger student model, or from the progressive expansion process itself. Together, these two ablations disentangle the effects of capacity growth, parameterization efficiency, and progressive adaptation in our framework. The results in Table 6 further validate the individual contributions of our core components. First, the necessity of progressive expansion is demonstrated by the poor performance

Table 6: The results of further ablation studies.

| Method | CoraFull-CL | | | Arxiv-CL | | |
|---|---|---|---|---|---|---|
| | AP/% ↑ | AF/% ↑ | H-Mean ↑ | AP/% ↑ | AF/% ↑ | H-Mean ↑ |
| CMLP w/ Non-expanding MLP | 10.82 | -81.38 | 13.69 | 11.23 | -72.33 | 15.98 |
| CMLP w/o Low-rank | 40.32 | -16.98 | 54.28 | 21.45 | -12.17 | 34.48 |
| CMLP | 42.66 | -14.54 | 56.91 | 23.48 | -9.53 | 37.28 |

Table 7: The impact of the different routing designs.

| Designs | CoraFull-CL | | | Reddit-CL | | |
|---|---|---|---|---|---|---|
| | AP/% ↑ | AF/% ↑ | H-Mean↑ | AP/% ↑ | AF/% ↑ | H-Mean↑ |
| Soft Gating | $20.63 \pm 0.61$ | $-67.32 \pm 1.98$ | 25.81 | $27.10 \pm 1.47$ | $-54.43 \pm 1.84$ | 34.92 |
| Hard Gating | $17.43 \pm 0.45$ | $-69.49 \pm 2.03$ | 21.69 | $26.78 \pm 1.03$ | $-62.57 \pm 2.25$ | 30.10 |
| **Energy-based Routing** | $42.66 \pm 1.32$ | $-14.54 \pm 0.47$ | 56.91 | $55.84 \pm 1.02$ | $-12.80 \pm 0.52$ | 66.73 |

of the Non-expanding MLP variant. The static MLP fails significantly on incremental tasks, with its AF dropping as low as -81.38% on CoraFull-CL, proving that task-specific capacity expansion is essential for mitigating catastrophic forgetting in graph class-incremental scenarios. Second, the effectiveness of low-rank decomposition is evident when comparing the full CMLP with the "w/o Low-rank" version. Beyond its primary goal of parameter efficiency, the low-rank constraint consistently yields higher H-Mean scores. This suggests that the low-rank structure acts as a beneficial regularizer, helping the model learn new class representations without excessively distorting the previously acquired knowledge base. Overall, CMLP achieves the best balance between plasticity and stability.

### E.3 Impact of the Different Routing Designs.

To further examine the routing mechanism, we compared several designs that differ in how experts are selected and combined. 1) Soft gating routing assigns continuous mixture weights to all experts through a learnable gating network, allowing each input to be processed by multiple experts simultaneously. 2) Hard gating routing, in contrast, selects only the top-1 expert based on gating probabilities. 3) Our energy-based test-time routing operates without a learned gating network: it dynamically selects experts according to their energy scores computed from input representations. We observe from Table 7 that our energy-based test-time routing achieves substantially better performance than both soft and hard gating strategies. The main reason is that gating-based routing requires labeled data for training at each time step; as a result, the gating network inevitably suffers from catastrophic forgetting and tends to overfit to the most recent classes, thereby degrading its ability to generalize across incremental tasks. In contrast, the energy-based approach performs routing directly at inference without requiring label supervision, making it more robust and stable under continual learning settings.

### E.4 Multi-Cycle Evaluation

To further evaluate the long-term stability of the proposed expand–distill–reset framework, we extended the original setting to a multi-cycle regime with 5 full GNN update cycles. The full label space was partitioned into 5 GNN cycles, each containing 14 classes. Within each GNN cycle, we further divided the 14 classes into 4 MLP update cycles, with class splits of 4/3/3/3, respectively. Therefore, the student MLP was updated four times before each teacher refresh, and after each GNN cycle, the accumulated student knowledge was distilled back into the teacher, followed by MLPs reset for the next cycle. This setting is designed to directly test whether the proposed asynchronous framework remains stable when the expand–distill–reset process is repeated multiple times over a substantially longer horizon than in the main experiments. The results in Figure 9 show that the framework remains stable across repeated GNN refreshes and does not collapse under longer continual operation. In particular, the GNN performance exhibits a gradual decline across cycles, which is expected because later cycles involve progressively more challenging continual adaptation as

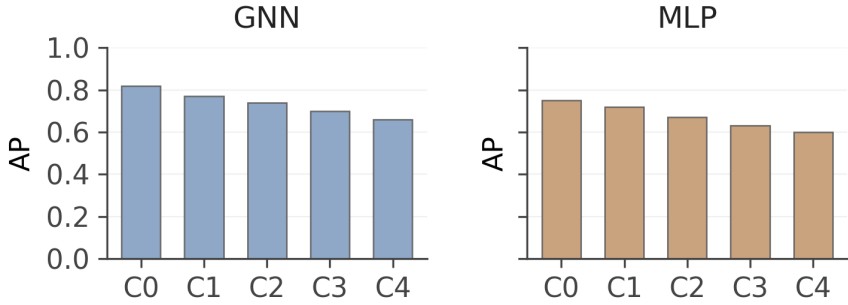

Figure 9: Performances of the GNN and MLP models across multiple GNN-MLP update cycles on CoraFull-CL.

the model accumulates more historical burden. In contrast, the reset MLP in each new GNN cycle is able to recover a relatively strong starting point after distillation, and then degrades more moderately within that cycle. This pattern suggests that the distill–reset mechanism is functioning as intended: the teacher consolidates knowledge across cycles, while the MLPs provide efficient short-horizon adaptation within each cycle. To quantitatively evaluate the necessity of the distill–reset mechanism in maintaining long-term stability, we conduct a comparative analysis between the standard CMLP and a variant without the reset operation (MLP w/o Reset) across five GNN-MLP update cycles. As summarized in Table 8, while both models initiate from the same performance level in C0, a clear performance divergence emerges as the number of cycles increases. Specifically, the MLP with the reset mechanism consistently maintains higher AP scores. The performance gap demonstrates that the reset operation effectively prevents the student model from being bogged down by the accumulation of historical, potentially redundant parameters. By periodically consolidating knowledge into the teacher and re-initializing the expansion process, CMLP ensures that the student retains high plasticity for new task adaptation without sacrificing the stability of previously acquired knowledge.

Table 8: Comparison of MLP performance (AP) with and without the reset mechanism across 5 GNN update cycles on CoraFull-CL.

| Variants | C0 | C1 | C2 | C3 | C4 |
|---|---|---|---|---|---|
| MLP w/ Reset | **75.57** | **72.40** | **67.62** | **63.50** | **60.81** |
| MLP w/o Reset | 75.57 | 71.32 | 65.48 | 60.27 | 56.49 |

### E.5 Inference and Training Efficiency Comparison

To demonstrate the efficiency of inference and training of our proposed method, we show comparisons between our method and several baseline methods. As shown in Figure 10, we observe that for inference efficiency, all methods except GraphSAGE and ERGNN adopt an MLP for inference and therefore achieve similarly low inference time. Although CMLP is based on the progressive mixture-of-experts, inference across these experts can be executed in parallel, so the overall inference time is primarily determined by the last expert. In contrast, GraphSAGE and ERGNN use a full GNN for inference, which results in significantly higher inference time. For training efficiency, the size of each solid circle represents the training time, where a larger circle indicates a longer duration. CMLP achieves the best AP performance with relatively low training cost. In comparison, NOSMOG and GLNN, which are based on GNN-to-MLP distillation, directly fine-tune the GNN and MLP to adapt to emerging graphs and thus require longer training time than the standard MLP. On the other hand, EWC and MAS adopt more complex strategies to address forgetting, such as parameter regularization, and consequently require more training time. This comparison highlights the effectiveness of CMLP in balancing inference and training efficiency.

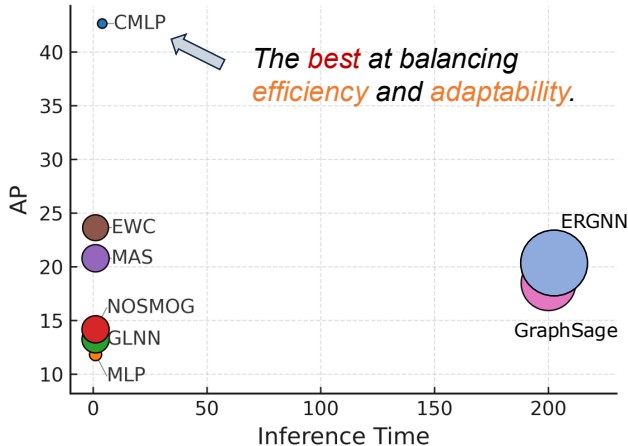

Figure 10: Inference and training efficiency comparison (the size of each solid circle represents the training time, where a larger circle denotes a longer duration).

# F  Further Discussion

## F.1  Distinction from Traditional Continual Learning

While both our setting and canonical graph continual learning operate on evolving graph structures, our problem formulation differs fundamentally in focus and objectives. Traditional methods primarily aim to mitigate catastrophic forgetting in GNNs by preserving performance on previously seen tasks. However, they overlook the need for efficient inference, which is critical in real-world, latency-sensitive applications. In contrast, our work focuses on enabling low-latency inference in class-incremental scenarios through GNN-to-MLP distillation, a direction that has not received attention in prior research. We propose a novel design paradigm that decouples the training schedules of the GNN and MLP. The GNN is updated at a lower frequency to reduce computational cost, while the lightweight MLP is frequently adapted to new data to support timely and accurate predictions. This means that when a new GNN update cycle begins, all previously expanded MLPs are discarded, and a new MLP is re-initialized from scratch with the original dimension $h$. In practice, a reasonable GNN update cycle can include 5 to 10 MLP update cycles, which limits unbounded memory growth by preventing the accumulation of the expanding set of MLP parameters. This asynchronous update strategy ensures both inference efficiency and adaptability, making it suitable for real-time, large-scale graph applications. Importantly, our approach provides a solution that is orthogonal to existing graph continual learning methods. As such, it can be seamlessly integrated with these methods to complement their strengths and broaden their applicability.

## F.2  Failure Analysis

A failure mode of the proposed energy-based routing mechanism is over-confident but incorrect expert selection, especially when the expert pool becomes larger or the class boundaries across incremental stages become less distinct. In these cases, the energy scores of multiple experts can become less separable, so the routing decision may be dominated by a small but misleading confidence difference, leading to misrouting between experts. This issue is also consistent with our empirical observation that performance degrades under longer MLP update cycles, where a larger number of experts makes confidence estimation less distinguishable. In our current setting, routing remains relatively stable partly because experts are trained on more weakly overlapping class groups. However, under more aggressive update regimes or less separated class distributions, expert assignment can become less reliable. Possible mitigation strategies include calibrating expert logits or energy scores to reduce over-confident routing, pruning or clustering experts to control redundancy as the expert pool grows, and introducing a fallback mechanism when the energy gap between the top-ranked experts is too small. These extensions could further improve routing robustness in longer and more challenging continual-learning settings.

### F.3 Future Work

In future work, we will explore lightweight expert clustering or pruning strategies to improve the scalability of the energy-based routing mechanism, particularly in scenarios where MLPs require high-frequency updates. Although our current design observes stable routing in practice, this is because each expert is trained on disjoint class groups. However, more aggressive update cycles may benefit from routing regularization or expert consolidation.

### Broader Impact

While the proposed framework is well suited to rapidly evolving graph learning settings, it also raises potential societal risks. A low-latency and continually adapting graph learning system could be misused for real-time surveillance or user profiling on social graphs. Moreover, biases present in evolving graph data may be preserved or even amplified through repeated GNN refresh and GNN-to-MLP distillation across update cycles. The routing mechanism may also fail by selecting an incorrect expert with high confidence, which could lead to unreliable predictions in sensitive applications. We hope future work will examine these risks more explicitly and develop appropriate safeguards.

