# OpenReview forum: "Bridging Efficiency and Adaptability: Continual Learning of MLPs on Class-Incremental Graphs"
_TMLR — Accepted by TMLR_

### Review · Reviewer_53ZZ · 2026-02-13

**Summary Of Contributions:**

Summary Of Contributions

This paper targets graph class-incremental learning on evolving graphs under a practical deployment constraint: graph-free, low-latency inference (i.e., using an MLP at test time). It proposes CMLP, an asynchronous framework that (i) updates a costly GNN teacher at low frequency, while (ii) frequently adapts a lightweight MLP via progressive expansion into a pool of MLP experts. At inference, the method uses energy-based test-time routing to select the most confident expert per node, and during periodic teacher updates it distills knowledge from the selected experts back into the GNN to mitigate forgetting.

Strengths
1. Clear problem framing: bridges continual adaptation with practical inference efficiency (graph-free / low-latency).
2. Simple, modular design: progressive experts + test-time routing avoids training a gating network that could itself forget.
3. Reasonable empirical coverage: multiple datasets, standard CL metrics (AP/AF/H-mean), comparisons to both continual-learning and GNN-to-MLP baselines, plus an efficiency discussion.

Weaknesses
1. Related work is incomplete for recent continual graph learning (CGL/GCIL), especially works on evolving domain shifts, expanding networks, and prompt/profiling-based GCIL (see Requested Changes).
2. Routing scalability remains a concern as the expert pool grows; energy separability and performance can degrade with more cycles.
3. The setting benefits from experts trained on disjoint class groups; robustness under class overlap / softer distribution shifts is not fully established.

**Audience:**

Yes

**Audience Explanation:**

The paper sits at the intersection of continual learning, graph ML, and efficient inference / distillation, which matches a substantial segment of TMLR’s readership. The proposed design pattern (fast lightweight continual adaptation + slower teacher refresh) is broadly relevant beyond the specific benchmarks.

**Broader Impact Concerns:**

No major ethical concerns are apparent.

**Claims And Evidence:**

Yes

**Claims Explanation:**

The main claims (asynchronous teacher/student updates, progressive expansion to reduce forgetting, energy-based test-time routing, and expert-guided distillation back to the GNN) are described concretely and supported by experiments with standard continual metrics (AP/AF/H-mean) across multiple datasets and baseline families. That said, broader claims about scalability are only partially validated, as performance/routing separability can drop when the number of experts (cycles) increases; additional analysis would strengthen confidence.

**Requested Changes:**

1.	Strengthen Related Work, with a natural discussion of recent continual graph learning that is close to this paper’s setting (evolving domain shifts, expanding networks, and prompt/profiling-based GCIL). In particular, clarify how these lines compare/contrast with the “GNN-to-MLP + asynchronous updates + routing” formulation, and whether/how they could be integrated.

[1] Qiao et al. GCAL: Adapting Graph Models to Evolving Domain Shifts. ICML 2025.

[2] Zhang et al. Topology-aware Embedding Memory for Continual Learning on Expanding Networks. KDD 2024.

[3] Niu et al. Replay-and-Forget-Free Graph Class-Incremental Learning: A Task Profiling and Prompting Approach. arXiv:2410.10341, 2024.

2.	Stress-test routing beyond the “clean” regime: since routing stability can benefit from experts trained on disjoint class groups, add at least one controlled experiment with class overlap / softer distribution drift to verify that energy-based confidence remains reliable.
3.	Make efficiency accounting more explicit: parameter growth per step, number of experts evaluated per query, and the practical cost of energy computation over experts (even if parallelized).
4.	Expand the analysis of the update schedule (teacher update frequency vs. student update frequency) and provide actionable guidance on stable operating regions.
5.	Add a brief failure-mode analysis (e.g., over-confident but wrong expert selection) and discuss mitigation (calibration, pruning/clustering, abstention).

---

> ### Author Response · Authors · 2026-03-12
> **Response to Reviewer 53ZZ - Part 1**
>
> We thank the reviewer 53ZZ for the constructive suggestions and respond to the comments as follows.
>
> **1. Related work on continual graph learning**
>
> Thanks for the suggestion. We agree that the related work section should better cover recent continual graph learning directions that are close in spirit but differ in problem formulation. **We have revised the extended related work in Appendix A.2**.
>
> **2. Routing scalability as the expert pool grows**
>
> We respectfully highlight that the routing scalability as the expert pool grows is reflected in our current analysis: when the MLP update cycle increases, AP decreases, and forgetting rises because the energy-based confidence scores become less distinguishable as more experts accumulate. At the same time, our method is explicitly designed to prevent unbounded expert growth: experts are only maintained within a fixed GNN update cycle, and all expanded MLP experts are discarded once the next GNN refresh begins, with their knowledge distilled back into the GNN. This bounds the number of active experts and limits memory growth in practice. Moreover, even within larger update cycles, CMLP remains more robust than strong baselines, outperforming NOSMOG and EWC when the cycle is fixed at 10, as shown in Figure 9. We have revised the manuscript to make this trade-off more explicit, including clearer guidance on stable update-cycle regimes and a discussion of scalability-oriented extensions such as expert pruning/clustering and routing regularization as promising future directions in Appendix F.4.
>
> **3. Robustness with class overlap**
>
> We agree that the current benchmarks are relatively clean in the sense that experts are trained on disjoint class groups, and this can make expert separation easier. In fact, we also note in the manuscript (F.4 Future work) that stable routing in the current design partly benefits from this property, and that more aggressive update regimes may require regularization or expert consolidation.
> At the same time, the current paper does provide evidence that the routing mechanism itself is meaningful beyond a trivial expert lookup: energy-based routing substantially outperforms both soft-gating and hard-gating routing variants in Table 10, suggesting that label-free confidence-based selection is more robust than learned gating under continual updates.
>
> To directly address the reviewer’s concern, **we have added a controlled stress test with class overlap by relaxing the disjointness between successive increments in Appendix E.2**. This allows us to test whether energy remains discriminative when task boundaries are less clean. We conduct the controlled experiment with class overlap on Arxiv-CL and Reddit-CL datasets, which consist of 40 classes, respectively. In the default setting, we divide them into five disjoint groups of 8 classes, corresponding to five MLP update cycles within one GNN update cycle. To stress-test routing beyond this clean regime, we additionally construct a controlled overlapping-class setting. The first cycle contains 8 classes, while each subsequent cycle contains 10 classes, including 8 newly introduced classes and 2 classes shared with the previous cycle. For these shared classes, we split the samples across adjacent cycles, assigning 70\% of the samples to the earlier cycle and the remaining 30\% to the later cycle. This design preserves the total of 40 unique classes while creating softer boundaries between adjacent experts, enabling us to evaluate whether energy-based routing remains reliable under class overlap. As shown in Table 8 in Appendix E.2, our proposed CMLP consistently outperforms the best EWC baseline across both datasets under the class-overlap setting. Notably, despite the 20\% class overlap between adjacent cycles, CMLP maintains a significantly lower forgetting rate (AF) compared to EWC. This demonstrates that our energy-based routing mechanism is robust to softer class boundaries, effectively assigning samples to the most relevant experts even when the distinction between tasks is not strictly disjoint.

---

> ### Author Response · Authors · 2026-03-12
> **Response to Reviewer 53ZZ - Part 2**
>
> **4. Efficiency**
>
> We thank the reviewer for the suggestion. The manuscript provides a parameter-complexity analysis of the progressive key-value expansion in Section 6, showing the base parameter count and the newly introduced parameters at each step, and highlighting that the added growth is structured and more compact than naïve expansion. Regarding inference, our current routing evaluates the active experts at test time by computing an energy score from each expert’s output logits and selecting the expert with the lowest energy. Thus, at time step $t$, the logical number of expert evaluations per query is the number of active experts in the current cycle. At the same time, the framework is designed for low-latency deployment: inference is performed through lightweight MLP experts rather than a full GNN, and the paper notes that expert inference can be executed in parallel, so the overhead is primarily determined by the slowest active expert, which is usually the most recent expert rather than a sequential sum over all experts.  Routing requires evaluating all available experts and computing one energy score per expert. Therefore, the total compute grows approximately linearly with the number of experts. For example, the inference time of the last expert is approximately 5× that of the first expert when there are five MLP update cycles. However, since energy computation itself is only a lightweight log-sum-exp reduction over logits, the dominant cost comes from the expert forward passes.
>
> **5. Update schedule**
>
> We thank the reviewer for this suggestion. Our framework is intentionally built on asynchronous updates, where the student MLP is updated more frequently for rapid adaptation, while the teacher GNN is refreshed at a lower frequency to incorporate structural information and provide a stable backbone for the next cycle. The current manuscript provides initial evidence on this trade-off: when the MLP update cycle increases from 3 to 5 to 7, AP decreases and forgetting increases, as shown in Table 4, because energy-based expert confidence becomes less distinguishable as more experts accumulate. At the same time, smaller cycles require more frequent GNN updates and therefore higher training costs. In particular, the current results suggest that shorter or moderate MLP-only phases provide the best balance between adaptation and retention, whereas overly long student-only phases lead to reduced energy separability and accumulated forgetting. We have added clearer guidance on the teacher/student update ratio in practice, emphasizing the trade-off between (i) frequent teacher refreshes for stronger structural consolidation and routing stability, and (ii) less frequent teacher refreshes for lower training cost and faster continual adaptation.
>
> **6. Failure analysis**
>
> We thank the reviewer for the suggestion. **We have added this failure analysis in Appendix F.3 and discussed practical mitigation strategies**. A failure case in our framework is over-confident but incorrect expert selection, especially when the expert pool grows, the update cycle becomes longer, or the class/distribution boundaries are less clean. In these cases, the energy scores of multiple experts may become less separable, making routing decisions less reliable. This concern is also consistent with our current discussion that stable routing in the present setting partly benefits from experts being trained on disjoint class groups, and that more aggressive update regimes may require additional regularization or expert consolidation.

---

### Review · Reviewer_VwsD · 2026-02-18

**Summary Of Contributions:**

This work explores the task of inference-efficient continual learning. To tackle the efficiency, GNN-to-MLP distillation is utilized. Key innovation includes an expansion strategy for a student MLP via a product of two low-rank matrices to minimize forgetting.
During the inference, the most confident student is selected amongst all cycles of expansion.

### Strengths

1. Clear motivation and practical relevance
The paper targets a realistic setting.
2. Progressive expansion idea seems like a resonable approach to avoid forgetting. It preserves prior knowledge without ruining old parameters.
3. Energy-based expert routing is allows to select experts without adding trainable parameters to a router, which theoretically makes routing more stable.
Besides, it allows quick expansion of an expert pool, which fits naturally with incremental a student expansion idea.

### Weaknesses
1. Insufficient ablation for the solution. The methodology combines MLP expansion, Mixture-of-experts with energy routing, bidirectional distillation and asynchronous schedule. This raises concerns about the utility of some components.
2. The description of the experimental setting is not clear, which make it hard to evaluate the contribution.
3. Catastrophic forgetting mitigation is partial, since the GNN still updates periodically and relies on distillation from a selected expert.

**Additional Comments:**

# Literature (updated 25/02/10

1. Zhang, Xikun, Dongjin Song, and Dacheng Tao. "Continual learning on graphs: Challenges, solutions, and opportunities." arXiv:2402.11565 (2024).
2. Rumiantsev, Pavel, and Coates, Mark. "Graph Knowledge Distillation to Mixture of Experts" TMLR (2024).
3. Veličković, Petar, et al. "Graph attention networks." ICLR (2018).
4. Li, Guohao, et al. "Training graph neural networks with 1000 layers." ICML (2021).
5. Zhang, Lei, et al. "DRGCN: Dynamic evolving initial residual for deep graph convolutional networks." AAAI (2023).

**Audience:**

Yes

**Audience Explanation:**

This paper is applying GNN-to-MLP distillation to the class-incremental scenarios. It presents a clearly novel approach of asynchronous update to solve it. While the task might be rather niche, the manuscript certainly will be interesting for people working on distillation or continual learning.

**Broader Impact Concerns:**

No ethical concerns

**Claims And Evidence:**

Yes

**Claims Explanation:**

1. &#x2705; "To the best of our knowledge, we are the first to investigate GNN-to-MLP distillation on class-incremental graphs, addressing the challenge of adapting to evolving graph data while maintaining inference efficiency." As far as i know, this is correct.
2. &#x2705; "We design an asynchronous update paradigm for GNN and MLPs to enable rapid adaptation to evolving graph data." It is presented in section 4.4. To the best of my knowledge, the asynchronous updates for GNN-to-MLP distillation is a novel idea.
3. &#x2705; "We propose a progressive MLP expansion strategy to mitigate forgetting, ensure low-latency inference, and support fast updates without access to past data." It is presented in section 4.2.
4. &#x2705; "We formulate the expanded MLPs as a mixture of experts to leverage knowledge from different stages and introduce an energy-based test-time routing mechanism to efficiently select the best expert for inference." This formulation is presented in section 4.3.
5. &#x2753; "We conduct extensive experiments on real-world graph datasets, demonstrating that our method effectively balances adaptability to new data and inference efficiency." The setup of the experiments is not clear from the text. Some baseline selection is odd. These factors render the result open to a range of interpretations.

**Requested Changes:**

### Critical

The experimental setup is unclear to me. Even after reading Appendix D. The original setups for distillation baselines were not designed for continual learning.
The continual learning baselines you use were designed for computer vision tasks, not graphs. How was each baseline adapted? What GNN were used for continual learning baselines? How were distillation baselines trained? How many update steps does a student do when a new graph is introduced?

Definition 1 does not specify relationships between nodes/edges of iteration t and iteration t+1. From Section 4, I can assume t+1 is an expansion of t, since you are using student expansion to retain knowledge. Could authors clarify on that?

I have a lot of questions regarding the expansion of a student. Its inclusion does not seem well-justified.
Why do you need it if each expanded MLP becomes an expert in PMoE anyway? You could have started with a fresh one each time. When expanding student MLP, why do you need a low-rank decomposition? Why are all experts discarded at the end of each cycle? If you read any of the papers on GNN-to-MLP distillation that you have cited in your work, the memory consumed by a teacher or student is tiny, while the real problem is always the latency of gathering all the required nodes.
Could the authors include some ablation study on the expansion to show it has some impact?
1. an experiment with expansion, but no low-rank decomposition;
2. another experiment without expansion, where a student MLP has a size the as after T expansions, yet from the beginning;
3. an experiment where old experts are not discarded.

Section 4.3 introduces an idea of Progressive Mixture of Experts. Could you discuss this idea with respect to a MoE-student introduced by Rumiantsev & Coates (2024)? You both seem to use MoE to increase the performance at the test time.

I cannot quite grasp the motivation behind selecting the continual learning baselines. The manuscript cites several papers on graph class-incremental learning, including a survey on the topic (Zhang et al., 2024). However, the continual learning baselines ignore all of them and instead select some older works unrelated to the graph domain. Could the authors explain this decision?

All experiments are done using GraphSAGE as a teacher (seemingly). Could you include experiments with better-quality teachers to show that your method generalizes? I ask to use GAT (Veličković et al., 2024), RevGNN (Li et al., 2021), and DRGAT (Zhang et al., 2023) as advanced teachers.

There are a lot of inconsistencies in the presented results. Figures 4 & 5 demonstrate that GLNN performs way better than your model, which contradicts Table 2 and the discussion in the text. Figure 6 shows that GNN performs better without the bidirectional update you are presenting, which fully contradicts the text and renders methods from section 4.4 to be actively harmful for the teacher performance (and thus for the student performance.)

The analysis in "Inference and Training Efficiency Comparison" makes very little sense. Inference time boost is achieved only because MLP is used instead on GNNs. So inference time is fully expected to be about the same amongst the distillation methods and significantly less that of a GNN teacher. The training time of your method is also fully expected to be less than the training of the disillation methods, because your method was specifically designed for this specific setting you use, while they are being somehow adapted. What are the training times of some lightweigted methods specifically designed for graph class-incremental learning?


### Minor

Page 1. "GNN-to-MLP distillation has been widely explored in *static* settings." Both works on distillation you are citing earlier on this page explore transductive and inductive settings. The latter simulates adding new nodes to the graph. Which of them are you referring to as "static"? Why not use the original terminology?

Page 4. In definition 2 everything after "Due to the computational overhead..." is an intuition behind your design choices. Consider moving it to the experimental setup section to increase the readability. Similar comment for the last sentences in definition 1.

From definition 3, is it correct that you update the teacher only 2 times during your experiments? Why exactly two?

---

> ### Author Response · Authors · 2026-03-12
> **Response to Reviewer VwsD - Part 1**
>
> We thank the reviewer VwsD for the constructive suggestions and respond to the comments as follows.
>
> **1. Experimental Setup**
>
> We thank the reviewer for raising this point. To clarify, the continual graph stream, task partition method, and evaluation protocol all follow the existing benchmarks: we use the four benchmark datasets from CGLB [1]. Each dataset is partitioned into five groups, the first group is used for the initial teacher training and distillation, and the remaining groups are introduced sequentially as incremental updates.
>
> **1.1 How was each baseline adapted?**
>
> We agree that the original distillation baselines were proposed for static graphs rather than continual graph learning. In our experiments, we adapt them to the same graph stream and incremental update protocol as our method so that all approaches are evaluated under the same evolving graph partitions and class-incremental process. Importantly, this adaptation should not be understood as claiming that the original distillation formulation is naturally suited to continual graphs. On the contrary, as illustrated in Figure 1(b), a direct continual adoption of standard GNN-to-MLP distillation would require repeatedly retraining the teacher GNN and re-distilling MLP whenever new graph data arrive, which is computationally expensive and delays adaptation. Moreover, Figure 2 highlights the two resulting issues: the deployed MLP remains stale until the teacher update/distillation finishes, and sequentially overwriting the student can lead to catastrophic forgetting of previously learned classes. Therefore, the purpose of including these distillation baselines is precisely to test how existing distillation methods behave when moved from their original static setting into the continual graph regime.
>
> Our continual learning baselines (GEM, MAS, LwF, EWC) were not used as graph continual learning methods on a GNN backbone. Instead, they were applied to the MLP side under our graph-free continual inference setting. This is consistent with the formulation in Figure 1(c), where the problem is cast as continual learning of MLPs under an evolving graph stream, rather than continual message-passing updates of a single GNN. Thus, these methods were adapted as standard continual learning regularization/replay strategies on the student MLP, while the graph stream itself still follows the CGLB benchmark. We have revised Appendix D.3 to describe this adaptation explicitly for each baseline.
>
> **1.2 What GNNs were used for continual learning baselines?**
>
> As we described in Appendix D.4, the teacher architecture in our overall framework is a basic GCN, and the student is a two-layer MLP. In addition, to examine whether the method depends on a specific teacher design, we also evaluated multiple teacher backbones in Appendix E.1 and reported the corresponding results in Figure 11.
>
> **1.3 Training of distillation baselines.**
>
> The distillation baselines were trained under the same CGLB continual graph stream and update protocol as our method. Specifically, the first graph partition is used to train the teacher GNN and perform the initial GNN-to-MLP distillation, and the remaining partitions are then introduced sequentially. Whenever a new graph partition arrives, the teacher–student distillation process is re-run using the emerging graph data. This training procedure corresponds to the continual distillation workflow illustrated in Figure 1(c), where the student model is repeatedly updated as the GNN updates. In this way, the original teacher–student distillation objective of each baseline is preserved, and the training is applied to the continual graph stream defined by CGLB rather than a static graph.
>
> **1.4 How many update steps does a student do when a new graph is introduced?**
>
> Each newly introduced graph partition corresponds to one incremental update event in the CGLB stream. In our framework, the student is updated at every such event, while the teacher is updated less frequently according to the asynchronous update cycle shown in Figure 1(c). Within each incremental event, the student model is trained until convergence on the current partition. This ensures that the student sufficiently adapts to the newly introduced graph data before the next incremental partition arrives.
>
> [1] Zhang, Xikun, Dongjin Song, and Dacheng Tao. "Cglb: Benchmark tasks for continual graph learning." Advances in Neural Information Processing Systems 35 (2022): 13006-13021.

---

> ### Author Response · Authors · 2026-03-12
> **Response to Reviewer VwsD - Part 2**
>
> **2. Definition 1**
>
> We would respectfully clarify that in our current formulation, $G_{t+1}$ is not assumed to be a graph expansion of $G_{t}$ in the sense that all nodes/edges of $G_{t}$ must be explicitly contained in $G_{t+1}$. Rather, $t$ and $t+1$ denote consecutive increments in the evolving graph stream following the previous benchmark, where each step introduces a new graph partition with a disjoint newly introduced class group for student-side updates. The “expansion” in Section 4 refers to student model expansion (i.e., progressive MLP expansion to preserve previous knowledge while adapting to new classes), not to a required superset relation between consecutive graph snapshots. We have revised Definition 1 in the revision for better clarity.
>
> **3. Connection with MoE-student introduced by Rumiantsev & Coates (2024)**
>
> We thank the reviewer for pointing out the relevant connection to Rumiantsev & Coates (2024). We agree that there is a high-level similarity: both their RbM student and our PMoE use an MoE-style student architecture to improve graph-free test-time inference over a single distilled MLP. In both cases, the motivation is to retain the low-latency advantage of MLP-style inference while improving student accuracy through expert specialization and routing.
>
> However, the two methods are designed for different problem settings and use MoE in different ways. Rumiantsev & Coates study a static GNN-to-MoE distillation setup for standard transductive/inductive node classification. Their student is a dedicated MoE architecture, Routing-by-Memory (RbM), where experts are trained jointly and specialization is explicitly encouraged through memory-based routing. Their goal is to improve student performance for a fixed graph distillation setting by replacing a single MLP with a better MoE student.
>
> In contrast, our PMoE is introduced in a class-incremental graph continual learning setting, where the central challenge is not only improving test-time accuracy, but also balancing adaptation to new graph partitions/classes and retention of prior knowledge across MLP updates. Accordingly, our experts are not jointly trained from scratch as a static MoE architecture. Instead, they arise from a progressive chain of expanded MLP students over time, and PMoE is formed by collecting these step-wise students within one update cycle. The role of PMoE is therefore tied to continual adaptation: it provides a pool of temporally specialized experts, and our energy-based routing selects the most suitable one at test time. This is different from RbM, where MoE is the student architecture itself, rather than a collection of progressively expanded continual students.
>
> A second key difference is the routing mechanism and its purpose. RbM performs routing by comparing hidden representations to expert-associated memory embeddings and uses additional losses to enforce representation-space clustering and balanced expert usage. By contrast, our routing is energy-based test-time expert selection over progressively expanded MLPs. We do not impose the same embedding-space specialization objective as RbM; instead, our routing is designed to decide which historical student is most reliable for the current sample in the continual class-incremental setting. In other words, RbM’s MoE is a static specialized student architecture, whereas PMoE is a continual-learning mechanism built on top of progressive student expansion.
>
> We clarified in the revision that the novelty of PMoE is not simply “using MoE for better test-time performance,” but using a progressive MoE over continually expanded students to address class-incremental graph adaptation and forgetting, which is outside the scope of the static RbM setting.
>
>
> **4. Teacher GNN models.**
>
> We thank the reviewer for this suggestion. Although the main experimental sections use a basic GCN for consistency, the current manuscript does not rely only on a single GNN teacher. In the appendix, we have evaluated the impact of different teacher GNN backbones, including GCN, GraphSAGE, GAT, and APPNP, across the four benchmark datasets. As reported in Figure 11, the performance differences among these teacher variants are relatively small, indicating that the final MLP performance is not strongly tied to a specific teacher architecture. This behavior is also consistent with the design of our framework. As described in Algorithm 1, GNN-to-MLP distillation is performed only at the beginning of each GNN update cycle; after that, the student evolves mainly through the proposed progressive expansion, PMoE routing, and expert-guided adaptation. Therefore, the final performance is influenced more by the student-side continual adaptation mechanism than by the exact choice of the initial teacher backbone. Importantly, our current results suggest that CMLP is largely teacher-agnostic, rather than being specialized to a single GNN teacher.

---

> ### Author Response · Authors · 2026-03-12
> **Response to Reviewer VwsD - Part 3**
>
> **5. Expansion of students**
>
> **5.1. Why not start with a fresh new MLP each time?**
>
> We thank the reviewer for this important question. The purpose of expansion is not merely to create multiple experts for PMoE, but to make each later student inherit and extend the knowledge learned in previous steps, instead of training a completely fresh MLP on each graph partition. In our setting, each expanded MLP is built upon the previously learned student parameters and then adapted to the newly arrived graph partition, so the student knowledge is accumulated progressively rather than reset at every step. This is important because learning from scratch at each step would lose the continuity of the previously acquired feature and decision structure.
>
> At the same time, although each later MLP accumulates prior knowledge, we cannot assume that the latest MLP is always the best model for all previously seen classes. After multiple expansions toward newer graph partitions, different MLPs may become better specialized for different subsets of classes or graph regions. This is precisely why we use PMoE: instead of heuristically deciding which step-specific MLP should be used for a given node, PMoE automatically selects the most suitable expert through routing. Therefore, expansion and PMoE play complementary roles: expansion preserves and accumulates student knowledge across partitions, while PMoE avoids forcing the latest student to serve as a universal expert for all old and new classes.
>
> **5.2. Why low-ranking decomposition?**
>
> We use a low-rank decomposition because the student is expanded repeatedly over time, and a naïve dense expansion would cause unnecessary parameter growth and eventually hurt inference efficiency. In our design, the newly added weights are generated by compact key-value matrices, while all previously learned weights remain fixed. This allows the student to add capacity for newly emerging graph partitions/classes without overwriting old knowledge, while keeping the expansion itself lightweight. The manuscript stated this motivation in Sec. 4.2: the key-value expansion is introduced to accumulate new knowledge with limited interference and to keep the MLP efficient for inference, rather than introducing more complex components that may increase latency.
>
> The low-rank decomposition is therefore mainly a parameter-efficient realization of progressive expansion. As analyzed in Sec. 6, instead of directly appending dense blocks, the key-value formulation introduces fewer parameters, which reduces parameter growth. The paper further showed that, if the key-value products are precomputed and cached after training, inference reduces to a standard forward pass, enabling progressive model growth with limited overhead and efficient, scalable inference. Thus, the purpose of low-rank decomposition is not compression for its own sake, but to make repeated student expansion compatible with our low-latency inference objective.
>
> **5.3. Why are all experts discarded at the end of each cycle?**
>
> We discard all experts at the end of each cycle because, in our framework, the PMoE is designed as a temporary within-cycle adaptation mechanism, whereas the refreshed GNN serves as the persistent long-term knowledge carrier across cycles. As shown in Fig. 3, the expanded MLP experts are created to handle the current sequence of graph partitions and to provide adaptive routing within the current update cycle. Once their knowledge has been used to guide the teacher update, the framework returns to the updated GNN for the next cycle, rather than maintaining an ever-growing pool of historical experts. In other words, experts are used to provide short-horizon specialization, while the GNN is responsible for consolidating this information and carrying it forward across cycles.
>
> There are two main reasons for this design. First, keeping all historical experts would continuously increase memory usage and routing complexity over cycles, which would conflict with our goal of maintaining lightweight, low-latency inference. Second, once a new GNN has been updated using the expert-guided process, retaining all previous experts becomes less necessary, because their useful knowledge is intended to be distilled back into the teacher. Therefore, discarding experts is not meant to ignore past knowledge; rather, it reflects our design choice that past knowledge should be consolidated into the updated GNN, instead of being stored indefinitely as an expanding expert bank.

---

> ### Author Response · Authors · 2026-03-12
> **Response to Reviewer VwsD - Part 4**
>
> **5.4. Ablation studies**
>
> Thanks for the suggestion. **We have added a further ablation study in Appendix E.3**. To better isolate the contribution of the proposed MLP expansion mechanism, we further consider two additional ablations. First, we evaluate a variant that retains progressive expansion but removes the low-rank decomposition, so that each expansion is implemented with standard full-parameter updates. Second, we evaluate a non-expanding MLP whose size is fixed from the beginning to match the final size obtained after expansions in our full model. This ablation controls for total model capacity and tests whether the gain comes merely from having a larger student model, or from the progressive expansion process itself. Together, these two ablations disentangle the effects of capacity growth, parameterization efficiency, and progressive adaptation in our framework.
>
> The results in Table 9 further validate the individual contributions of our core components. First, the necessity of progressive expansion is demonstrated by the poor performance of the Non-expanding MLP variant. The static MLP fails significantly on incremental tasks, with its AF dropping on CoraFull-CL, proving that task-specific capacity expansion is essential for mitigating catastrophic forgetting in graph class-incremental scenarios. Second, the effectiveness of low-rank decomposition is evident when comparing the full CMLP with the ``w/o Low-rank'' version. Beyond its primary goal of parameter efficiency, the low-rank constraint consistently yields higher H-Mean scores. This suggests that the low-rank structure acts as a beneficial regularizer, helping the model learn new class representations without excessively distorting the previously acquired knowledge base. Overall, CMLP achieves the best balance between plasticity and stability.
>
> | Method                    | CoraFull-CL AP/% ↑ | CoraFull-CL AF/% ↑ | CoraFull-CL H-Mean ↑ | Arxiv-CL AP/% ↑ | Arxiv-CL AF/% ↑ | Arxiv-CL H-Mean ↑ |
> | ------------------------- | ------------------ | ------------------ | -------------------- | --------------- | --------------- | ----------------- |
> | CMLP w/ Non-expanding MLP | 10.82              | -81.38             | 13.69                | 11.23           | -72.33          | 15.98             |
> | CMLP w/o Low-rank         | 40.32              | -16.98             | 54.28                | 21.45           | -12.17          | 34.48             |
> | CMLP                      | 42.66              | -14.54             | 56.91                | 23.48           | -9.53           | 37.28             |
>
>
> Regarding the experiment where old experts are not discarded, in our framework, even if the old experts are not physically discarded, they are no longer used after the Expert-guided GNN Updating stage. This is because their knowledge has already been consolidated into the updated teacher GNN during this step. In the next cycle, the student MLP is reinitialized and trained based on the refreshed teacher, rather than directly relying on the previously learned experts. Therefore, keeping the old experts would not affect the learning process or inference behavior of the framework, as they are unused once the teacher has been updated.

---

> ### Author Response · Authors · 2026-03-12
> **Response to Reviewer VwsD - Part 5**
>
> **6. Baselines**
>
> Our choice of continual learning baselines was driven by the target model and deployment setting of this work, rather than by the graph domain alone. Specifically, our goal is not continual learning of a single GNN, but continual learning of distilled MLP students for graph-free, low-latency inference. As stated in the paper, existing graph class-incremental methods mainly focus on updating a single GNN with sufficient capacity to retain past knowledge, whereas our setting studies how to continually adapt an MLP student after GNN-to-MLP distillation.
>
> For this reason, we selected generic continual learning baselines such as GEM, MAS, LwF, and EWC because they are architecture-agnostic and can be applied directly to the evolving student MLP under the same update schedule as our method. The paper evaluates all baselines under the same evolving graphs and update protocol as CMLP, so these methods serve as controlled references for whether standard continual-learning strategies are sufficient for the student side of our problem.
>
> To address the reviewer's concern, **we have added ERGNN as an additional baseline in Table 2**. To adapt ERGNN to our setting, we follow its experience replay strategy by selecting representative nodes from previous steps and storing them in a replay buffer when updating the teacher GNN. The updated teacher is then distilled into the student MLP for graph-free inference. As shown in Table 2, our method CMLP still outperforms ERGNN. It is worth noting that ERGNN leverages stored nodes from previous steps through replay, which provides additional historical information beyond what is available in our setting. In contrast, CMLP does not access previous graph data during student updates, yet it still achieves stronger performance, demonstrating the effectiveness of the proposed framework.
>
> Meanwhile, we highlight that graph class-incremental methods do not fully match our deployment setting. Our goal is to balance adaptability and efficiency, whereas graph class-incremental methods require updating the GNN model at every MLP cycle, which effectively corresponds to the continual distillation setting illustrated in Figure 1(b). Such frequent GNN updates are computationally expensive, as message passing over large-scale graphs incurs significant overhead, making fast adaptation impractical. Also, the deployed MLP remains static until the ongoing GNN update is completed, failing to rapidly produce reliable inferences on newly emerging node classes.
>
> **7.  Inconsistencies in the presented results.**
>
> We believe there is a misunderstanding in the interpretation of Figures 4–6. These figures do not contradict Table 2 or the text.
>
> First, Figures 4 and 5 do not report the same aggregate metric as Table 2. Table 2 reports the overall continual-learning results (e.g., AP / AF / H-Mean) across the full class-incremental process, whereas Figures 4 and 5 visualize the performance matrix M, where each entry $M_{i,j}$ denotes the classification accuracy on the node classes introduced at step $j$, after training on the graph at step $i$. As stated in the manuscript, these heatmaps are intended to show how performance evolves over time, especially the retention of previously learned classes. The conclusion from these figures is exactly that CMLP retains prior knowledge more effectively than GLNN and EWC, while GLNN focuses more on the current graph and suffers much stronger forgetting. This is fully consistent with both Table 2 and the discussion in the text.
>
> Second, Figure 6 does not show that the bidirectional/expert-guided update is harmful. The comparison there is among three different teacher states: GNN at $t_0$​ (the initial teacher before a GNN update cycle), GNN w/o Expert (the updated teacher after MLP update cycles but without guidance from the MLP experts), and GNN w/ Expert (the updated teacher with PMoE guidance). The purpose of Figure 6 is to show that the teacher itself undergoes forgetting after a full GNN update cycle, i.e., both updated variants perform worse than the original t_0 teacher on the old classes from stage t_0. However, among the two updated teachers, the paper explicitly states that the expert-guided teacher mitigates this forgetting relative to the unguided teacher. Therefore, Figure 6 supports our claim that expert-guided GNN updating is beneficial to mitigate the forgetting issue.
>
> To avoid this confusion, we revised the captions of Figures 4–6 to make the comparison targets more explicit:
> (1) Figures 4–5 show class-wise retention patterns, not the same summary metric as Table 2; and
> (2) Figure 6 compares two updated-teacher variants after one cycle, where PMoE guidance reduces forgetting relative to the unguided update, even though both remain below the original pre-update teacher on old classes.

---

> ### Author Response · Authors · 2026-03-12
> **Response to Reviewer VwsD - Part 6**
>
> **8. Inference and Training Efficiency Comparison**
>
> We thank the reviewer for this comment. We agree with the main observation that the inference-time advantage in Fig. 10 is largely due to using an MLP student instead of a GNN. In fact, this is already consistent with our intended setting: the goal of GNN-to-MLP distillation is precisely to enable graph-free, low-latency inference, and the manuscript explicitly notes that “all methods except GraphSAGE adopt an MLP for inference and therefore achieve similarly low inference time.” Thus, Fig. 10 should not be interpreted as claiming a surprising inference-time improvement of CMLP over other MLP-based distillation baselines; rather, it shows that our continual adaptation mechanism preserves the same low-latency inference regime while improving performance.
>
> The intent of Fig. 10 was not to claim that training-time differences are unexpected or universally favorable to CMLP, but to show that, under the same evolving-graph protocol and update schedule used in our experiments, CMLP achieves stronger accuracy with a relatively moderate training cost. This point is grounded in the way the baselines were instantiated in the paper: all compared methods were evaluated under the same evolving graphs and update schedule as CMLP, and the text explicitly explains that methods such as NOSMOG/GLNN require direct fine-tuning both GNN and MLP during adaptation, while EWC/MAS introduce additional continual-learning overhead through parameter regularization. In contrast, our design decouples the GNN and MLP schedules, updating the GNN less frequently and adapting the lightweight MLP more often, which is exactly the deployment setting we target.
>
> Regarding the reviewer’s question about lightweight methods specifically designed for graph class-incremental learning, we agree that this is an important perspective. Our current baseline set emphasizes methods that are directly comparable to our target problem, rather than methods that continue to rely on a graph model at test time. As discussed in the paper, most existing graph class-incremental methods focus on updating a single GNN and often rely on replay buffers, subgraph memory, prompting, or other graph-specific mechanisms. These are highly relevant to the broader graph continual incremental learning literature, but they address a different deployment objective from ours, since our primary goal is to maintain low-latency MLP-based inference after distillation.
>
> **9. The static settings**
>
> We thank the reviewer for this comment. We agree that the cited GNN-to-MLP distillation works consider both transductive and inductive settings. Here, the use of “static” was not meant to contrast with that terminology, but rather with continual learning. In other words, those prior works study distillation in a fixed/non-continual setting, whereas our work considers a class-incremental continual graph setting in which the teacher–student learning process evolves over time. We clarified this point in the revision to avoid ambiguity.
>
> **10. Definitions 1 and 2.**
>
> Thanks for the suggestion. We moved these explanatory parts out of the definitions and placed them in the surrounding experimental setup, while keeping Definitions 1 and 2 more concise and purely formal.

---

> ### Author Response · Authors · 2026-03-12
> **Response to Reviewer VwsD - Part 7**
>
> **11. Teacher updates only 2 times**
>
> In the current experimental setup, the teacher GNN is updated only twice. This was a deliberate design choice because our main focus in the paper is continual learning on the MLP student side, where the lightweight student is updated more frequently while the teacher is refreshed less often in the asynchronous framework. In other words, the current experiments emphasize whether the student can continually adapt over multiple incoming graph partitions before the next teacher refresh.
>
> To address the reviewer’s concern, **we have added a multi-cycle evaluation in Appendix E.5**. We extended the original setting to a multi-cycle regime with 5 full GNN update cycles. The full label space was partitioned into 5 GNN cycles, each containing 14 classes. Within each GNN cycle, we further divided the 14 classes into 4 MLP update cycles, with class splits of 4/3/3/3, respectively. Therefore, the student MLP was updated four times before each teacher refresh, and after each GNN cycle, the accumulated student knowledge was distilled back into the teacher, followed by MLPs reset for the next cycle. This setting is designed to directly test whether the proposed asynchronous framework remains stable when the expand–distill–reset process is repeated multiple times over a substantially longer horizon than in the main experiments. The results in Figure 12 show that the framework remains stable across repeated GNN refreshes and does not collapse under longer continual operation. In particular, the GNN performance exhibits a gradual decline across cycles, which is expected because later cycles involve progressively continual adaptation as the model accumulates more historical burden. In contrast, the reset MLP in each new GNN cycle is able to recover a relatively strong starting point after distillation, and then degrades more moderately within that cycle. This pattern suggests that the distill–reset mechanism is functioning as intended: the teacher consolidates knowledge across cycles, while the MLPs provide efficient short-horizon adaptation within each cycle. Overall, the multi-cycle experiment helps verify that the proposed framework is not tied to only two teacher updates and can generalize to longer asynchronous continual-learning processes.

---

> > ### Comment · Reviewer_VwsD · 2026-03-22
> > **Response to Authors**
> >
> > Thank you for the updated manuscript and clarifying notes. Currently, my questions are fully addressed for
> > - Experimental Setup
> > - Teacher GNN
> > - Definition 1 & 2
> > - New baseline
> > - Figure 6
> >
> >
> > ### 3. Connection with MoE-student
> >
> > I agree with your analysis of RbM and PMoE. They both designed for different settings that dictated the design. However, both of them are build on the same idea of selecting the most suitable expert. That allows PMoE to preserve knowledge on the graph stream and RbM to perform in transductive setting.
> >
> >
> > ### 5. Expansion of students
> >
> > I thank the authors for the expanded ablation study. The design choices seem clear. Still, I kindly request to make an ablation experiment where old experts are kept. Simply keep experts between cycles. Consider line 5 of Algorithm 1. It states to collect MLPs from the current cycle. Instead, collect MLPs from all the cycles to ensure knowledge are retained in PMoE. The rest of the algorithm can be kept the same.
> >
> >
> >
> > ### 7. Inconsistencies in the presented results.
> >
> > Look at the Figure 4 and Figure 5. The current captions state "Performance heatmaps over time..." and the legends include a colormap to the right of each figure. Darker colors on the colormap indicate higher performance. GLNN has darker colors, hence it has to perform better according to your figure. This is inconsistent with (a) caption, that states your model is better; (b) text of section 5.2; (c) potentially, table 2.
> > From the text: "CMLP demonstrates a stronger ability to retain past knowledge and achieves superior
> > classification accuracy on previously learned classes compared to GLNN and EWC". I can not see it in these figures in the current state.
> >
> > It could be due to my misinterpretation of those figures. Therefore, I ask authors to fix it.
> > 1. Update captions to indicate what "performance" is shown.
> > 2. Update the text with some guidance on how to read this figure properly. Specify what the term "lower-triangular performance" means.
> >
> >
> > ### 8. Inference and Training Efficiency Comparison
> >
> > I agree with the authors' perspective on inference. Yes, a preservation of the low-latency inference and a moderate training cost are important perspectives to consider. Please, add one or two points of graph class-incremental learning papers on Fig 10. This will considerably strengthen the position of your approach. Any graph class-incremental learning method with the public source code from the Related Work section will suffice.

---

> > > ### Author Response · Authors · 2026-03-27
> > > **Response to Reviewer VwsD**
> > >
> > > We appreciate Reviewer VwsD's further comments and suggestions. Following our previous clarifications, we respond to the remaining comments as follows.
> > >
> > > **1. Connection with MoE-student**
> > >
> > > Thanks for the comment. We agree with the reviewer that both RbM and our PMoE share the same high-level idea of selecting the most suitable expert at test time. We have explicitly highlighted and discussed this connection with RbM in the related work in Appendix A.1  and clarified that while both methods rely on expert selection, RbM is a static MoE student for graph distillation, whereas PMoE is a progressive MoE formed under a class-incremental continual learning setting, where experts correspond to different stages of the evolving graph stream.
> > >
> > > **2. Expansion of students**
> > >
> > > Thanks for the suggestion. We have conducted an additional ablation where experts are retained across cycles, i.e., instead of resetting MLPs after each GNN update cycle, we accumulate all experts from previous cycles and perform routing over the full set, and have provided in E.5 (Table 11). Specifically, the MLP with the reset mechanism consistently maintains higher AP scores. The performance gap demonstrates that the reset operation effectively prevents the student model from being bogged down by the accumulation of historical, potentially redundant parameters. By periodically consolidating knowledge into the teacher and re-initializing the expansion process, CMLP ensures that the student retains high plasticity for new task adaptation without sacrificing the stability of previously acquired knowledge. This aligns with our goal of equipping CMLP with high short-term plasticity, allowing it to adapt quickly to evolving graph streams before the next teacher refresh.
> > >
> > > **3. Inconsistencies in the presented results.**
> > >
> > > Thanks for the suggestion. We have revised the captions and the corresponding text. In the current figure 4, each entry $M_{i,j}$ in each subfigure denotes the classification accuracy on classes introduced at time step $j$, evaluated after training up to time step $i$. The diagonal entries ($i=j$) reflect performance on newly learned classes, while the lower-triangular region ($i>j$) reflects performance on previously learned classes (knowledge retention). Stronger lower-triangular values indicate better retention of past knowledge.
> > >
> > > **4. Inference and Training Efficiency Comparison**
> > >
> > > Thanks for the suggestion, we have added ERGNN as a graph class-incremental learning method in Figure 10 (now as Figure 9). ERGNN and GraphSage have comparable training and inference overheads.
> > >
> > > All changes in the revised manuscript are highlighted in red. Thank you again for your valuable feedback and suggestions.

---

> > > > ### Comment · Reviewer_VwsD · 2026-03-28
> > > > **Response to Authors**
> > > >
> > > > Thank you for your hard work on updates and new experiments. All my comments were fully addressed.

---

> > > > > ### Author Response · Authors · 2026-03-28
> > > > > **Response to Reviewer VwsD**
> > > > >
> > > > > We are glad that these concerns have been addressed. We appreciate your time and valuable feedback!

---

### Review · Reviewer_EzeN · 2026-02-26

**Summary Of Contributions:**

This paper addresses how to perform GNN-to-MLP knowledge distillation in a class-incremental graph setting, where new node classes emerge over time and low-latency inference is required. The authors propose CMLP, built on four main ideas:

1. **Asynchronous GNN/MLP update paradigm:** The GNN is updated infrequently (expensive), while lightweight MLPs are updated at each time step (cheap), decoupling structural learning from fast adaptation.
2. **Progressive MLP expansion via key-value decomposition:** At each new time step, the MLP's hidden layers are expanded using low-rank key-value matrices while freezing previously learned parameters, preventing catastrophic forgetting by construction.
3. **Energy-based test-time routing:** The expanded MLPs are treated as a mixture of experts, and at test time the expert with the lowest energy score is selected — no learned gating network needed.
4. **Expert-guided GNN updating:** When the GNN is periodically retrained, the accumulated MLP experts serve as teachers via knowledge distillation back into the GNN, mitigating GNN-side forgetting.

Experiments on four benchmarks (CoraFull-CL, Arxiv-CL, Reddit-CL, Products-CL) show large improvements over both GNN-to-MLP distillation and continual learning baselines.

## Key Strengths

- **Novel problem formulation.** Clearly identifies a real gap: existing GNN-to-MLP distillation assumes static graphs, and existing graph continual learning does not target low-latency MLP inference.
- **Simple, principled design.** Progressive expansion with frozen old parameters eliminates forgetting by construction. Energy-based routing avoids a learned gating network that would itself be subject to forgetting.
- **Strong empirical gains.** Substantial improvements over baselines (e.g., AP of 42.66 vs. 23.64 on CoraFull-CL). Includes ablations, sensitivity analyses, routing comparisons, and efficiency comparisons.

## Key Weaknesses

- **Limited scale of incremental steps.** All experiments use only 5 MLP update cycles within one GNN update cycle. Performance degrades with more cycles, and it remains unclear how the method handles significantly more steps (e.g., 20–50).
- **Monotonically growing model size.** Within a cycle, the number of experts and total parameters grow linearly. No concrete memory usage numbers or practical limits are discussed.
- **Narrow continual learning baselines.** The baselines (EWC, MAS, LwF, GEM) are relatively dated. More recent graph-specific continual learning approaches (ERGNN, SSM, CaT) are discussed but not compared against.
- **Single GNN update cycle evaluated.** Long-term behavior over repeated cycles of "expand MLPs → distill back into GNN → reset MLPs" is not empirically demonstrated.
- **Incomplete sentence on page 3 (Minor)** — clear editing oversight.

**Audience:**

Yes

**Audience Explanation:**

The paper sits at the intersection of three active research areas: GNN-to-MLP distillation, graph continual learning, and efficient inference on dynamic graphs. The problem formulation, enabling low-latency inference on graphs with evolving class distributions, is practically relevant for recommendation systems, social networks, and knowledge graphs. The asynchronous update paradigm is a sensible design principle that could inspire future work beyond this specific instantiation. Researchers working on scalable graph learning, continual learning, and knowledge distillation would likely find the framing and results of interest.

**Broader Impact Concerns:**

The paper lacks a Broader Impact Statement. Key concerns: (1) the low-latency, continually adapting node classification framework could facilitate real-time surveillance or user profiling on social graphs; (2) the asynchronous GNN-MLP distillation loop may propagate or amplify biases in graph data across update cycles without any fairness mechanism; (3) no discussion of failure modes when the routing mechanism selects wrong experts in safety-critical deployments. A brief acknowledgment of these risks would be appropriate.

**Claims And Evidence:**

Yes

**Claims Explanation:**

**Yes, with minor reservations.**

The central claims — that CMLP achieves superior performance on class-incremental graphs by combining progressive expansion with energy-based routing — are well supported. Improvements over baselines are large and consistent across four datasets, with statistical significance reported (paired t-test, p < 0.05). The ablation study (Table 3) convincingly isolates the contributions of expansion and mixture-of-experts, and the routing comparison (Table 5) clearly demonstrates the advantage of energy-based routing over learned gating alternatives.

However, the evidence could be strengthened in several areas:

- The evaluation is limited to a single GNN update cycle with at most 5–10 MLP steps. Claims about "long-term knowledge retention" would benefit from multi-cycle experiments over extended time horizons.
- The efficiency comparison (Figure 10) does not report exact numbers, making precise comparison difficult.
- AP scores, while much better than baselines, remain low in absolute terms (e.g., ~23% on Arxiv-CL, ~25% on Products-CL). Some discussion contextualizing these numbers (e.g., upper bounds from joint training) would be helpful.
- Variance/standard deviations are not reported despite experiments being repeated three times.

**Requested Changes:**

1. **Multi-cycle evaluation:** Run experiments over 3–5 full GNN update cycles to validate long-term stability of the expand-distill-reset loop.
2. **Stronger baselines:** Add at least one recent graph continual learning method (e.g., CaT, ERGNN) adapted to the MLP inference setting.
3. **Report standard deviations:** Include confidence intervals in all tables given the three repeated runs.
4. **Oracle upper bound:** Report jointly trained model performance to contextualize the absolute AP scores.
5. **Memory tracking:** Report peak memory usage and parameter counts at each time step.
6. **Proofreading:** Fix the incomplete sentence on page 3 and do a thorough editing pass.
7. **Routing failure analysis:** Provide examples or analysis of when energy-based routing misroutes between experts.

---

> ### Author Response · Authors · 2026-03-12
> **Response to Reviewer EzeN - Part 1**
>
> We thank the reviewer EzeN for the constructive suggestions and respond to the comments as follows.
>
> **1. Limited scale of incremental steps.**
>
> We would like to clarify that, in our framework, the MLP update horizon is not intended to grow arbitrarily long within a single GNN update cycle. The purpose of our asynchronous design is precisely to balance frequent lightweight MLP updates with less frequent but structurally informed GNN refreshes, rather than to treat the MLP as a standalone continual learner over a very long horizon. As described in Sec. 4.4 and Appendix F.1, the GNN is updated at a larger time scale because GNN training is more expensive, while the MLP is updated more frequently to provide timely graph-free inference in between GNN refreshes. A reasonable GNN cycle therefore contains a moderate number of MLP updates (e.g., 5–10), after which the accumulated student knowledge is distilled back into the GNN and the MLP is reinitialized for the next cycle.
>
> From this perspective, very long MLP-only horizons such as 20–50 updates within one GNN cycle are not the primary operating regime targeted by our method, because they would weaken the intended balance between short-term adaptability and periodic structural consolidation. In fact, the paper already shows that larger MLP cycle counts increase forgetting in Table 4 and Figure 9, which is exactly why the cycle ratio must be chosen carefully rather than pushed indefinitely. Our claim is therefore not that CMLP is designed for arbitrarily long MLP-only continual updates, but that it provides a better tradeoff between adaptation and retention under a moderate asynchronous cycle ratio, which is the regime motivated by our problem setting. To make this clearer, we revised the manuscript to emphasize that the key design is to balance MLP and GNN updates, not how to maximize the number of MLP updates within a single GNN cycle.
>
> **2. Multi-cycle evaluation**
>
> Thanks for the suggestion. **We have added a multi-cycle evaluation in Appendix E.5**. We extended the original setting to a multi-cycle regime with 5 full GNN update cycles. The full label space was partitioned into 5 GNN cycles, each containing 14 classes. Within each GNN cycle, we further divided the 14 classes into 4 MLP update cycles, with class splits of 4/3/3/3, respectively. Therefore, the student MLP was updated four times before each teacher refresh, and after each GNN cycle, the accumulated student knowledge was distilled back into the teacher, followed by MLPs reset for the next cycle. This setting is designed to directly test whether the proposed asynchronous framework remains stable when the expand–distill–reset process is repeated multiple times over a substantially longer horizon than in the main experiments. The results in Figure 12 show that the framework remains stable across repeated GNN refreshes and does not collapse under longer continual operation. In particular, the GNN performance exhibits a gradual decline across cycles, which is expected because later cycles involve progressively continual adaptation as the model accumulates more historical burden. In contrast, the reset MLP in each new GNN cycle is able to recover a relatively strong starting point after distillation, and then degrades more moderately within that cycle. This pattern suggests that the distill–reset mechanism is functioning as intended: the teacher consolidates knowledge across cycles, while the MLPs provide efficient short-horizon adaptation within each cycle.

---

> ### Author Response · Authors · 2026-03-12
> **Response to Reviewer EzeN - Part 2**
>
> **3. Memory tracking**
>
> Thanks for the suggestion. We agree that, within one GNN update cycle, the number of experts and the total student parameters increase monotonically as the MLP is progressively expanded. This is an intentional tradeoff of our design. However, the growth is substantially moderated by the key-value low-rank expansion, which was introduced exactly to keep the student lightweight and inference-efficient under repeated updates. As stated in Sec. 4.2, the expansion is designed to “ensure that the MLP remains efficient for inference” and to maintain “a lightweight structure suitable for real-time large-scale applications.” Sec. 6 further shows that, instead of directly appending dense blocks, the added parameter cost per step is reduced to a compact low-rank form, and the resulting overhead scales linearly with the expansion size.
>
> Following the setup in Appendix D.4, **we have provided a table of the step-wise memory footprint of progressively expanded MLPs in Section F.2**. Table 11 shows that, under our intended operating regime, the practical memory footprint of PMoE remains modest. More importantly, our method is not designed to let the expert pool grow indefinitely. As clarified in the revised manuscript, the key design is to balance MLP and GNN updates, rather than to maximize the number of MLP updates within a single GNN cycle. After a moderate number of MLP updates, the accumulated student knowledge is distilled back into the GNN, and the next cycle begins.
>
> **4. Stronger baselines**
>
> We thank the reviewer for the suggestion. **We have added ERGNN as an additional baseline in Table 2**. To adapt ERGNN to our setting, we follow its experience replay strategy by selecting representative nodes from previous steps and storing them in a replay buffer when updating the teacher GNN. The updated teacher is then distilled into the student MLP for graph-free inference. As shown in Table 2, our method CMLP still outperforms ERGNN. It is worth noting that ERGNN leverages stored nodes from previous steps through replay, which provides additional historical information beyond what is available in our setting. In contrast, CMLP does not access previous graph data during student updates, yet it still achieves stronger performance, demonstrating the effectiveness of the proposed framework.
>
> Meanwhile, we note that ERGNN does not fully match our deployment setting. Our goal is to balance adaptability and efficiency, whereas ERGNN requires updating the GNN model at every MLP cycle, which effectively corresponds to the continual distillation setting illustrated in Figure 1. Such frequent GNN updates are computationally expensive, as message passing over large-scale graphs incurs significant overhead, making fast adaptation impractical. Also, the deployed MLP remains static until the ongoing GNN update is completed, failing to rapidly produce reliable inferences on newly emerging node classes.
>
> **5. The efficiency comparison**
>
> The absolute inference time of the last expert in CMLP is around 0.5 ms when the number of cycles is 5, while other graph-to-MLP distillation methods have inference times of around 0.1 ms. Since the difference is still on the millisecond scale, the additional overhead remains very small in practice. **We have revised Section 6 for better clarity**.
>
> **6. Oracle upper bound**
>
> We thank the reviewer for the suggestion. **We have added the performance of the jointly trained model as an oracle upper bound in Table 2**.
>
> **7. standard deviations**
>
> We thank the reviewer for the suggestion. We have added the standard deviations in Tables 6 and 7 as an extended version, since Table 2 does not have sufficient space to include them. Similarly, we have added the standard deviations in Tables 3 and 4.

---

> ### Author Response · Authors · 2026-03-12
> **Response to Reviewer EzeN - Part 3**
>
> **8. Routing failure analysis.**
>
> We thank the reviewer for the suggestion. A failure mode of the proposed energy-based routing mechanism is over-confident but incorrect expert selection, especially when the expert pool becomes larger or the class boundaries across incremental stages become less distinct. In these cases, the energy scores of multiple experts can become less separable, so the routing decision may be dominated by a small but misleading confidence difference, leading to misrouting between experts. This issue is also consistent with our empirical observation that performance degrades under longer MLP update cycles, where a larger number of experts makes confidence estimation less distinguishable. In our current setting, routing remains relatively stable partly because experts are trained on more weakly overlapping class groups. However, under more aggressive update regimes or less separated class distributions, expert assignment can become less reliable. Possible mitigation strategies include calibrating expert logits or energy scores to reduce over-confident routing, pruning or clustering experts to control redundancy as the expert pool grows, and introducing a fallback mechanism when the energy gap between the top-ranked experts is too small. These extensions could further improve routing robustness in longer and more challenging continual-learning settings. **We have added the failure analysis in Appendix F.4**.
>
> **9. Proofreading**
>
> We thank the reviewer for pointing this out. We have revised this incomplete sentence and done thorough editing in the new version.
>
> **10. Broader Impact**
>
> We thank the reviewer for this important comment. We added a Broader Impact Statement discussing potential risks and limitations of the proposed framework in the Appendix.

---

> > ### Comment · Reviewer_EzeN · 2026-03-29
> >
> > Dear Authors,
> >
> > Thank you for your responses to my concerns. They have been fully addressed.
> >
> > Best regards,
> > Reviewer EzeN

---

### Author Response · Authors · 2026-03-16
**Summary of Revisions in Response to Reviewer Comments**

We sincerely thank all reviewers for their thoughtful and constructive feedback on our work. We are grateful for the time and effort each reviewer has dedicated to evaluating our paper. Based on the insightful comments, we have carefully revised the manuscript to improve its clarity, technical rigor, and presentation.

In the revised version, we have addressed the key points raised by the reviewers. Specifically, we have:

1. **Added a multi-cycle evaluation** to examine the long-term stability of the proposed asynchronous framework. The new experiments simulated multiple GNN update cycles and verified that the expand–distill–reset process remains stable over extended continual learning horizons (Appendix E.5).

2. **Provided detailed memory footprint analysis** for progressively expanded MLP experts, demonstrating that the proposed low-rank expansion maintains practical memory usage within each update cycle (Appendix F.2).

3. **Included an additional baseline from graph continual learning**, adding ERGNN to the comparison. The method is adapted to our setting by applying experience replay during teacher updates, followed by distillation to the student MLP (Table 2).

4. **Reported standard deviations** for experimental results in additional tables in the appendix (Tables 6 and 7).

5. **Added oracle upper-bound results** using jointly trained models to contextualize the performance under full supervision (Table 2).

6. **Added a failure-mode analysis**, discussing cases where energy-based routing may incorrectly select experts and outlining potential mitigation strategies (Appendix F.4).

7. **Added a broader impact discussion**, addressing potential societal risks (Appendix).

8. **Clarified the experimental setup**, including the adaptation of baselines and the student–teacher update schedule under the CGLB benchmark (Appendix D.3).

9. **Revised three definitions** to clarify the relationship between consecutive graph snapshots and enhance the conceptual clarity of the definitions.

10. **Expanded the experimental section with additional ablation studies**, including variants without low-rank decomposition and without progressive expansion to isolate the contribution of the proposed student expansion mechanism (Appendix E.3).

11. **Improved the discussion of efficiency comparisons**, clarifying that the inference-time advantage mainly arises from graph-free MLP inference and that the proposed framework maintains this efficiency while improving continual adaptation of MLP.

12. **Expanded the related work discussion** to include recent continual graph learning methods and clarify the differences between our asynchronous distillation framework and existing GCIL approaches.

13. **Added a controlled stress test with class overlap** by relaxing the disjointness between successive increments to test whether energy remains discriminative when task boundaries are less clean (Appendix E.2).

All changes in the revised manuscript are highlighted in red for easy identification. We have also provided detailed justifications in our point-by-point responses.

Thank you again for your valuable feedback and support.

---

### Decision · Action_Editor_1yJp · 2026-04-21

**Recommendation:** Accept as is

**Audience:**

Yes

**Audience Explanation:**

The paper presents a novel task and problem formulation that is likely to be of interest to members of the graph learning research community.

**Claims And Evidence:**

Yes

**Claims Explanation:**

The paper makes the following claims:

(C1) The paper investigates GNN-to-MLP distillation on class-incremental graphs and proposes an asynchronous update paradigm to enable rapid adaptation to evolving graph data.

(C2) The paper presents an MLP expansion strategy to mitigate forgetting. This also ensures low-latency inference and supports fast updates. The expansion strategy is formulated as a mixture of experts with an energy-based test-time routing mechanism.

(C3) The paper presents results on real-world graph datasets demonstrating the strengths of the method in terms of adapting to new data and performing efficient inference.

Claims C1 and C2 are supported in the paper through a clear methodology and principled design. Claim C3 is supported via thorough experimentation and a careful ablation study.